

# Physics-based simulation of hydrological processes in a high-elevation glaciated environment focusing on groundwater

Xinyang Fan[1,2], Florentin Hofmeister[3,4], Bettina Schaefli[1], and Gabriele Chiogna[2]

[1]Institute of Geography and Oeschger Centre for Climate Change Research, University of Bern, Bern, Switzerland
[2]Department of Geography and Geosciences, GeoZentrum Nordbayern, Friedrich-Alexander-University Erlangen-Nuremberg, Erlangen, Germany
[3]Chair of Hydrology and River Basin Management, School of Engineering and Design, Technical University of Munich, Munich, Germany
[4]Bavarian Academy of Sciences and Humanities, Munich, Germany

**Correspondence:** Xinyang Fan (xinyang.fan@unibe.ch)

**Abstract.** Understanding the role of groundwater is crucial to improving the quantification of the hydrological response to climate change in high-elevation glaciated environments. However, few studies have been conducted due to the lack of in-situ hydroclimatic observations, the complex topography, and the difficulty of characterizing surface-subsurface water exchange processes in these terrains. In this study, we adopt a fully-distributed, physics-based hydrological model, *WaSiM*, with an inte-
grated 2-dimensional groundwater module to quantify the observed streamflow variations and their interactions with groundwater in a high-elevation glaciated catchment (Martell Valley) in the central European Alps since the 2000s. Extensive field observations (meteorology, vegetation, glacier mass balance, soil properties, groundwater levels, river discharge) are collected to analyze hydrological processes and to constrain the model parameters. We observe that shallow alpine groundwater levels respond nearly as quickly as streamflow to snowmelt and heavy rainfall inputs, as their measured hydrographs show. Be-
cause hydrological models rarely simulate this quick groundwater response, this highlights the need for improved subsurface parametrization in hydrological modeling. Surprisingly, subsurface lateral flow plays a minor role in river discharge generation at the study site, providing new insights into the hydrological processes in such an environment. Lastly, our results underline the challenges of integrating point-scale groundwater observations into a distributed hydrological model, with important implications for future piezometer installation in the field. This study sheds new light on surface-subsurface hydrological processes
in high-elevation glaciated environments. It highlights the importance of improving subsurface representation in hydrological modeling.

## 1   Introduction

With the rising temperature and increasing frequency and intensity of extreme hydrological events, such as floods and droughts, many regions have experienced severe adverse impacts of climate change globally (IPCC, 2021). Alpine glaciers are among
the most sensitive landscapes to climate change, with widespread retreat and decreasing snow precipitation observed globally (Rounce et al., 2023; Matiu et al., 2021; Marcolini et al., 2017). This trend is expected to persist, threatening water availability





for downstream communities and ecosystems (Drenkhan et al., 2023), disrupting hydropower operations (Tobias et al., 2023; Schaefli et al., 2019), and increasing the flooding risks from glacier lake outbursts (Banerjee et al., 2024). It is crucial to quantify and understand the hydrological processes and water storage changes in high-elevation alpine environments for making
improved projections of future water resource changes and their associated social and ecological impacts (van Tiel et al., 2024).

Many studies have attempted to quantify the impacts of climate change on high alpine water resources by using so-called conceptual hydrological models. These models represent the integrated effect of runoff generation and transfer to the stream network with simplified mathematical approaches, particularly linear reservoirs (Horton et al., 2022; Fatichi et al., 2015;
Dobler and Pappenberger, 2013; Schaefli et al., 2005). In such models, the hydrological dynamics in the subsurface, groundwater recharge, and baseflow generation are often overly simplified and poorly represented, thus adding high uncertainties to future hydrological projections (Puspitarini et al., 2020; Tuo et al., 2018). These models typically have few parameters and are relatively easy to calibrate on the observed streamflow. The resulting apparently high simulation performance should, however, not obscure the fact that high alpine streamflow simulations do typically not capture well the winter baseflow (e.g., Fig.3 in the
work of Michel et al. (2022), Fig.5 in the work of Schaefli and Huss (2011)) nor the onset of melt-induced streamflow increases (e.g., Fig.4 in the work of Schaefli et al. (2014), Fig.5 in the work of Schaefli and Huss (2011)).

Improving streamflow simulation in high alpine regions has been a central topic among mountain hydrologists (Somers and McKenzie, 2020; Beniston et al., 2018; Fayad et al., 2017; Freudiger et al., 2017). A prevailing assumption is that inaccuracies
in spatial snow distribution are the primary cause of melt-induced streamflow timing and magnitude errors, leading to calls for improved snow models (Freudiger et al., 2017; Warscher et al., 2013; Dobler and Pappenberger, 2013). However, existing snow models often perform satisfactorily when evaluated against snow station measurements and remote sensing data (Hofmeister et al., 2022; Carletti et al., 2022; Schlögl et al., 2016). This raises the question of whether the insufficient discharge simulation stems from other missing processes or structural elements in the models, particularly the overly simplified representation of
groundwater (Somers and McKenzie, 2020). Recent conceptual modeling studies indicate that incorporating groundwater observations for calibration has limited impact on improving discharge simulations, but aids in enhancing parameter identifiability in low-elevation catchments (<1000 m a.s.l.) with little snow influence (Pelletier and Andréassian, 2022). However, the role of groundwater in hydrological modeling in high-elevation catchments and the broader interactions between the cryosphere and groundwater remain largely unexplored (van Tiel et al., 2024).


The limited understanding of surface-subsurface processes in high alpine catchments is largely attributed to sparse in-situ observations, incomplete hydroclimatic datasets (Hofmeister et al., 2023a), and the heterogeneous nature of subsurface hydraulic properties (Müller et al., 2024; Chen et al., 2023; Fan et al., 2023a). Quantifying the interactions between groundwater and surface water is highly challenging even without snowmelt and glacier melt impacts (Abhervé et al., 2023; Fan et al.,
2023b; Bonotto et al., 2022; Krause et al., 2007). In mountainous regions, complex topography, diverse geology, and the interplay of surface-subsurface hydrological processes at high elevations further complicate modeling efforts. Several studies



have explored these interactions, including investigations of groundwater capacity in storing and releasing water during high and low streamflow periods (Müller et al., 2024), the impacts of geology on groundwater-surface water dynamics (Arnoux et al., 2021; Somers et al., 2016), and the application of physics-based models to simulate surface-subsurface processes (Eng-
dahl, 2024; Chen et al., 2023; Thornton et al., 2022). However, challenges remain. For instance, Chen et al. (2023) found that semi-distributed models lack sufficient spatial resolution to capture groundwater response to hydrological inputs (e.g., peak snowmelt, glacier recession, rainfall) and interactions with river networks. Thornton et al. (2022) demonstrated the difficulty of coupling the fully-distributed surface and groundwater models externally and calibrating them to the groundwater level observations. Engdahl (2024) noted the absence of uncertainty quantification in most coupled models due to high computational
demands. Still, they emphasized that the projected relative changes in watersheds with this coupling approach remain meaningful. Overall, advancing our understanding of groundwater-surface water interactions in high alpine environments requires overcoming significant observational and modeling challenges, with progress contingent on improving data availability and quality, refining modeling frameworks and observational networks, and computational strategies.

Here we aim to address the challenges of understanding and simulating interactions between cryosphere and groundwater by employing a fully-distributed, physics-based hydrological modeling approach. Our research questions are: (i) Can the alpine surface-subsurface processes be modeled with an integrated physics-based model alone, as opposed to the external coupling framework? (ii) How to set up a fully-distributed physics-based hydrological model, incorporating point-scale piezometer observations, to reproduce streamflow in such an environment? The research focuses on the Martell Valley in the central European
Alps (Northern Italy), a well-documented and accessible site for field data collection. We adopt the Water Balance Simulation Model (*WaSiM*) (Schulla, 2024, 1997), which integrates key hydrological processes in high-elevation glaciated catchments (e.g. snow and glacier dynamics) and has an embedded groundwater model that avoids the need for external model coupling. This study is among the first to develop and implement an integrated physics-based modeling framework that simulates surface-subsurface processes in the high alpine glaciated environment (Thornton et al., 2022).


## 2  Study site

The Martell Valley is a high-elevation glaciated catchment in the central European Alps (Northern Italy, see Fig. 1) with an elevation range of 1840 to 3760 m a.s.l and complex fine-scale topography (Hofmeister et al., 2022). The catchment has an area of 77 km$^2$, closed at the Zufritt Lake. The catchment is covered by around 40% of bare rocks, 34% of grassland (growing
upon shallow soil formation), 9% of forests, and 17% of glaciers (CORINE Land Cover, 2018). The glaciers are located in the southwest of the catchment, and the largest are Langenferner, Zufallferner, and Fürkeleferner. These glaciers reached their largest extension during the last Little Ice Age (LIA); they started showing a retreating trend in the 1850s, and the retreat has speeded up since the 1980s (Knoflach et al., 2021). Glacier retreat is one of the key drivers causing the high-alpine geosystem changes, such as land cover and landscape, ecological succession, hydrology, and geomorphology (Ramskogler et al., 2023;



Knoflach et al., 2021).

Geologically, the catchment is dominated by metamorphic minerals and rocks, such as orthogneiss, phyllite, quartz, and marble, with spatially heterogeneous soil formation (Ramskogler et al., 2023; Berra et al., 2012). Climatically, the catchment (mean altitude: 2800 m a.s.l.) has a mean annual temperature of -1.2°C between 2007 and 2023, with a maximum of 0.2°C in

2022 and a minimum of -2.9°C in 2010 (estimated with the interpolated catchment mean, see Section 3.1). The mean annual total precipitation and modeled evapotranspiration (see Section 3.1) of the catchment are 1048 mm yr$^{-1}$ and 167 mm yr$^{-1}$ for the same period. 2014 is the wettest year, with the highest annual precipitation of 1368 mm and the lowest evapotranspiration of 122 mm. 2022 is the driest year, with the lowest annual precipitation of 661 mm and the highest annual evapotranspiration of 218 mm. This catchment is relatively dry among the European Alps (Höge et al., 2023; Ceperley et al., 2020; Isotta et al.,

2018). Note that the role of groundwater could be different for the catchments with annual precipitation of more than 2000 mm, for instance, as compared to a relatively dry catchment quantified here.





**Figure 1.** Map of the study site - Martell Valley (a) in the central European Alps, and (b) its zoomed-in map with the observed meteorological and hydrological stations. In (a), the dark shadow denotes the European Alps (Jarvis et al., 2008) and the red dot denotes the location of the catchment. In (b), the numbers in black refer to the hydrological station IDs listed in Table 2, and the numbers (S1-S9) in violet refer to the subcatchment IDs. Triangles indicate the locations of meteorological stations and squares of hydrological stations. The glacier cover in 2000 originates from the glacier inventory (Knoflach et al., 2021). The background is the Digital Elevation Model (DEM) data of this region (di Bolzano Alto Adige, 2006).





## 3   Observed temporal and spatial data

### 3.1   Meteorological data

The observed daily meteorological station data are used as inputs in the *WaSiM* hydrological model. There are 12 climate stations in and around the Martell Valley with records of air temperature, precipitation, solar radiation, relative humidity, wind speed, and snow height from 1972 to 2024. The list of the stations is given in Table 1 and their locations are shown in Figure 1. The data at Station Langenferner are requested from the Institute of Atmospheric and Cryospheric Sciences of the University of Innsbruck in Austria. The Station Careser is operated by Meteo Trentino in Italy (http://storico.meteotrentino.it/web.htm?
ppbm=T0065&rs&1&df). Other station data are obtained from the Autonomous Province of Bozen (Bolzano) - South Tyrol in Italy. The elevations of all observed stations vary from 1720 to 3328 m a.s.l. The Zufritt Station has the longest daily climate records among all stations. Hofmeister et al. (2023a) assessed the data quality and corrected the observation errors of these climate stations until 2020. We apply the same procedure to the extended time series between 2020 and 2024.

The spatial and temporal interpolation of the climate station data to the retained spatial grid (25m × 25m) is performed in the *WaSiM* modeling environment. *WaSiM* supports irregularly observed data (i.e. missing values) and uses the following interpolation methods: the solar radiation, relative humidity, and precipitation recorded at the observed stations are spatially interpolated at daily time steps with the Inverse Distance Weighting (IDW) method. The wind speed and air temperature are interpolated with a combined method of IDW and elevation-dependent regression. The potential evapotranspiration (PET) is
calculated with the Penman-Monteith method (Brutsaert, 1982; Monteith, 1981), which is available in *WaSiM*, by using the daily temperature, relative humidity, solar radiation, and wind speed.

The snow height (SH) data are measured at the Zufritt and Zufallhuette stations (see Tab. 1). We transform the snow height to the snow water equivalent (SWE) with the ΔSnow model (v1.0.2), which is available in the R package "*nixmass*" (Winkler
et al., 2021). This model adopts a process-based multi-layer method and considers the relationship between snow density (Pistocchi, 2016) and predefined regional parameters for Italy (Guyennon et al., 2019) in the SWE calculation. The SWE data are then used to calibrate the parameters of the snow module in *WaSiM* (Details on snow module calibration are given in Section 4.1).



**Table 1.** Meteorological stations with daily records in and around the Martell Valley. Abbreviations: mean air temperature (T in [°C]), precipitation (P in [mm]), solar radiation (GS in W m$^{-2}$), relative humidity (LF in %), wind speed (WG in [m s$^{-1}$]), snow height (SH in [cm]).

| Station name | Station ID | Longitude [°] | Latitude [°] | Elevation [m a.s.l.] | Start date | End date | Climate variables |
|---|---|---|---|---|---|---|---|
| Schoentaufspitze | 06040WS | 10.6286 | 46.5029 | 3328 | 2003.01.01 | 2024.01.15 | T, WG |
| Madritsch | 06090SF | 10.6144 | 46.4938 | 2825 | 2003.01.01 | 2024.01.15 | T, P, GS, LF, WG |
| Sulden | 06400MS | 10.5953 | 46.5159 | 1907 | 2003.01.01 | 2024.01.15 | T, P, GS, LF |
| Hintermartell | 11400MS | 10.7269 | 46.5169 | 1720 | 2009.01.01 | 2024.01.15 | T, P, GS, LF, WG |
| Weissbrunnspitz | 24170WS | 10.7740 | 46.4940 | 3253 | 2004.01.01 | 2024.01.15 | T, P, GS, LF, WG |
| Rossbaenke | 24300SF | 10.8194 | 46.4693 | 2255 | 2002.01.01 | 2024.01.15 | T, LF, WG |
| Weissbrunn | 24400MS | 10.8318 | 46.4868 | 1900 | 2002.01.01 | 2024.01.15 | T, LF |
| Careser | T0065 | 10.6991 | 46.4226 | 2600 | 1990.10.01 | 2024.01.15 | T, P, LF, WG |
| Ghiacciaio | T0473 | 10.7183 | 46.4513 | 3093 | 1990.12.01 | 2024.01.15 | T, LF, WG |
| Langenferner | - | 10.6139 | 46.4725 | 2967 | 2021.10.02 | 2023.10.02 | T, GS, LF, WG |
| Zufritt | 11200BM | 10.7251 | 46.5090 | 1851 | 1972.01.01 | 2024.01.15 | T, P, SH |
| Zufallhuette | 16ZH | 10.6781 | 46.4814 | 2265 | 2004.12.01 | 2023.02.23 | SH |

## 3.2 Hydrological data

We installed five groundwater piezometers in the river floodplains, which are relatively flat with the pre-assumed existence of groundwater storage. Furthermore, we installed two river gauging stations during 2019 and 2023 (see Fig. 1 and Tab. 2). The groundwater piezometers are located at elevations above 2300 m a.s.l., and the river gauges cover an elevation range of 1878 to 2347 m a.s.l. The groundwater heads are measured at hourly intervals. The river water levels are measured at 15-min intervals. The river water levels are then transformed to the discharge [m$^3$ s$^{-1}$] with the rating curves derived in the work of Hofmeister et al. (2023b). The main river, named Plima, has its source at the Langenferner, flows through the catchment, and enters the Zufritt Lake in the northeast. There is an official discharge station (Plima Station, see location in Fig. 1 and Tab. 2) installed by the Autonomous Province of Bozen (Bolzano) - South Tyrol in the center of the catchment, where the data are measured at 10-min intervals. We transform all hydrological data into daily time steps for the hydrological model simulations. Note that due to the extreme flooding events between 12 July and 9 August 2023, the river gauge ID 450499 was completely destroyed. The measured data of this station cannot be used after the flooding event as the rating curve has significantly changed.

We quantify the trends of the time-series hydrological variables with the non-parametric Mann-Kendall test (Kendall, 1955; Mann, 1945) and Sen's slope method (Sen, 1968). The MATLAB function *mannkendall* (v1.0.0) (Collaud Coen and Vogt, 2021; Collaud Coen et al., 2020) is adopted to first remove the first-order autocorrelation in the data with a 3PW pre-whitening



procedure and then to quantify the trends.

**Table 2.** Observed groundwater piezometers and river gauging stations in the Martell Valley. The Plima Station is an official station operated by the Autonomous Province of Bozen (Bolzano) - South Tyrol. The river gauge ID 450499 was destroyed in the extreme flooding events between 12 July and 9 August 2023.

| Variable | Station ID | Longitude [°] | Latitude [°] | Elevation [m a.s.l] | Start date | End date |
|----------|-----------|---------------|--------------|---------------------|------------|----------|
| Groundwater | 4478 | 10.65186 | 46.46949 | 2454 | 2022.11.24 | 2023.11.01 |
| Groundwater | 4479 | 10.65302 | 46.47014 | 2455 | 2022.11.24 | 2023.11.01 |
| Groundwater | 4476 | 10.65507 | 46.46828 | 2447 | 2023.05.12 | 2023.11.01 |
| Groundwater | 4473 | 10.66646 | 46.47441 | 2333 | 2023.03.29 | 2023.11.01 |
| Groundwater | 4672 | 10.67205 | 46.47382 | 2319 | 2023.09.20 | 2023.11.01 |
| River | 450346 | 10.66730 | 46.47256 | 2347 | 2019.07.17 | 2023.11.01 |
| River | 450499 | 10.70838 | 46.49637 | 1878 | 2020.06.30 | 2023.08.07 |
| River | Plima | 10.67680 | 46.47630 | 2311 | 2014.06.01 | 2024.01.15 |

## 3.3 Spatial data

The Digital Elevation Model (DEM) data of the study site are provided by the Province of Bolzano (di Bolzano Alto Adige, 2006) and available in a spatial resolution of 0.5 m × 0.5 m in 2006. The soil profile and attributes in 2018 are extracted from the Harmonized World Soil Database (version 2.0) in a 1 km × 1 km resolution (FAO & IIASA, 2023; Ramskogler et al., 2023) (see Fig.A1a). The land use and land cover data of the site in 2018 are obtained from the Corine Land Cover Data Center (CORINE Land Cover, 2018) in a resolution of 100 m × 100 m (see Fig.A1b). The geological information of the site is obtained from the map produced by Berra et al. (2012). The glacier coverage maps of the site are obtained from Knoflach et al. (2021) and are available in the years 1850, 1911, 1959, 1985, 1997, 2005, 2013, 2017, and 2019. We calculate the fractional glacier-covered area (ranging between 0 and 1) with these data and use them to calibrate the glacier module in *WaSiM*. Annual glacier mass balance observations are only available for the Langenferner between 2004 and 2021 (Galos et al., 2017; Autonome Province of Bolzalno, 2007), which are used to evaluate glacier module performance as well. All spatial raster data are resampled to 25 m × 25 m with ArcGIS Pro (v3.1.0) for the hydrological modeling.

Using the spatial data of local slope and flow accumulation, which are generated with the add-on software of *WaSiM* named *Tanalys* based on the DEM data, we calculate the Topographic Wetness Index (TWI) of the catchment for each grid cell to understand the topographic control of the hydrological processes by using Eq. 1 (Pourali et al., 2016; Beven and Kirkby, 1979):

$$\text{TWI} := \ln\left(\frac{a}{\tan\beta}\right), \tag{1}$$



where $a$ [m$^2$] is the local upslope catchment area per unit contour length along the flow pathway; $\tan\beta$ is the local steepest

downslope in radians by comparing the slope of the modeled grid cell with that of its neighboring cells.

## 4    Integrated hydrological modeling

We use the Water Balance Simulation Model (*WaSiM*) (Schulla, 2024, 1997), version 10.08.00. Different versions of this model

have previously been used in many studies, such as for investigating fast local infiltration (Bronstert et al., 2023), snow process

and redistribution (Förster et al., 2018; Warscher et al., 2013), land use changes impact on streamflow (Cornelissen et al.,

2013), and flash floods in the Mediterranean catchments (Garambois et al., 2013), but to our knowledge not in comparable

with high-elevation settings. Thornton et al. (2022, 2021) used the *WaSiM* snow module but coupled it with *HydroGeoSphere*

to represent the subsurface in the alpine site. Here we adopt the whole *WaSiM* to integrally model the surface-subsurface pro-

cesses in the high-elevation glaciated catchment.

*WaSiM* is a 2-dimensional, fully-distributed, and modular-based hydrological model. It simulates hydrological partitioning

and fluxes on a regular horizontal grid, with the stream network being represented as selected grid cells of the domain. The

subsurface has a pre-defined number of soil and groundwater layers (for numerical reasons, the same bottom depth for both

domains) (Schulla, 2024). Lateral exchange between grid cells is not simulated for surface and unsaturated subsurface pro-

cesses, except for those grid cells that are adjacent to stream network cells. For groundwater, the groundwater flow equation

considering Darcy's law is implemented to simulate lateral groundwater flow (Schulla, 2024).

Besides meteorological time series (air temperature, precipitation, solar radiation, wind speed, relative humidity), *WaSiM*

also requires spatially gridded inputs, such as local slope, aspect, flow direction, flow accumulation, and river networks. These

raster files are generated with *Tanalys*, which takes the DEM data as input and is an associated software of *WaSiM*. Each mod-

ule in *WaSiM* can be flexibly (de-)activated based on the site characteristics. A detailed description of each module is available

in the works of Schulla (2024, 1997). Here, we focus on describing the key modules for setting up the hydrological model for

the Martell Valley including the snow, glacier, unsaturated zone, and groundwater modules to simulate the surface-subsurface

hydrological processes in the catchment.

### 4.1    Snow and glacier modules

In *WaSiM* (version 10.08.00, Schulla (2024)) the snowfall is simulated with a threshold approach and a user-defined temperature-

transition range (parameters "TRS" and "Ttrans" in Table 3). Liquid and solid precipitation interception is calculated with

classic bucket approach (Schulla, 2024; Förster et al., 2018; Kopp, 2017). Due to the relatively low share of forest in this

catchment (9% forest cover), snow interception is not considered in the model setup. For snow melt, we adopt the advanced





energy-balance method together with the lateral wind-driven and gravitational snow redistribution algorithms (Schulla, 2024; Thornton et al., 2021; Shrestha et al., 2014; Warscher et al., 2013; Gruber, 2007). The outflow from the snowpack infiltrates into the soil, which is solved with the 1-dimensional vertical Richards equation (Richards, 1931). The infiltration amount is the minimum between the fillable porosity of the uppermost soil layer and the total amount of rainfall and melt water per time step. The surface runoff is generated by the infiltration excess and saturation excess of the soil moisture.

We adopt the dynamic glacier module (Schulla, 2024) to simulate the glacier area evolution by using a volume-area scaling approach (Bahr et al., 1997; Chen and Ohmura, 1990). In the glacier module, the glacier mass gain (due to snow accumulation) and melt (due to snow, firn, and ice melt) are calculated over each year in *WaSiM*. The transformation from snow to firn takes one year if the snow does not melt within one season, and from snow to ice takes seven years (Schulla, 2024). The snow-on-ice melt is simulated using the same energy-balance approach as adopted in the Snow Module. The firn and ice melt are simulated with an extended temperature-index approach (Hock, 1999, 1998) that includes global radiation information at each daily time step on each glaciated cell (see Eqs.2.21.2 and 2.21.3 in the work of Schulla (2024)). The melts from snow-on-ice, firn, and ice are transformed into the streamflow conceptually. Each melt component is handled with a separate linear reservoir in *WaSiM*. Each reservoir has a specific storage coefficient, thus leading to different retention times. This is a simplified assumption of the drainage system that evolves on the glacier during each melting period (Schulla, 2024).

## 4.2 Unsaturated zone and groundwater model

All grid cells that are not stream grid cells contain a fixed number of unsaturated and saturated domain layers, which is set to five in total here (this applies also to glacier grid cells). Input to the unsaturated zone module is the sum of non-intercepted rainfall and melt water from snow and ice (if present). Infiltration and vertical fluxes between the five unsaturated soil layers are solved iteratively with the 1-dimensional Richards equations vertically (Schulla, 2024, 1997; Richards, 1931). The properties for each soil layer and for each soil type are defined per grid cell, such as soil thickness, vertical hydraulic conductivity, and characteristics of the macropores (Schulla, 2024).

To simulate the groundwater head changes, the unsaturated zone is coupled with a horizontal 2-dimensional groundwater model. The bottom depth of the aquifer is required (for numerical reasons) to be identical to the bottom depth of the soil module in *WaSiM* (Schulla, 2024). The shallow aquifer is modeled as unconfined, given that the observed depths to the water table (DTW) are mostly within 1 m. The water is only allowed to flow vertically in the unsaturated zone but can flow horizontally in the saturated zone. The unsaturated zone module first computes the vertical flux (i.e. the percolation) from the lowest unsaturated soil layer to the groundwater. This flux is then taken as the upper boundary condition for the groundwater model to calculate the lateral fluxes, which are solved with the Darcy equation. The net changes of the groundwater table are converted into a vertical flux, which is given back to the lowest unsaturated soil layer as inflow or outflow (Schulla, 2024). The aquifer properties are defined as spatially homogeneous. Five properties are calibrated in the groundwater model (see Tab.4), including





the horizontal hydraulic conductivities, aquifer thickness, storage coefficient, and colmation of the soil, which is a term used in *WaSiM*: it controls the water exchange rate between the adjacent saturated soil cells and the stream network cells.

## 4.3 Streamflow generation

The generation of streamflow based on water mobilized at the level of grid cells follows two stages: First, the surface runoff and the subsurface lateral flow (so-called "direct flow" and "interflow" in the model) are computed per grid cell; they are then transformed separately to the stream by a single reservoir cascade (isochrone approach with additional retention in each class) (Schulla, 2024). The surface runoff is generated when saturation excess of soil moisture or Hortonian infiltration excess occurs. The subsurface lateral flow for each layer per grid cell is generated based on the water content of the soil layer, hydraulic con-

ductivity, and local slope. Baseflow contribution to streamflow is obtained via direct lateral exchange of adjacent saturated soil cells to stream cells. These stream grid cells are generated with the add-on *Tanalys* software of *WaSiM* based on the DEM data and do not necessarily correspond to actual observed stream locations. The average values of each runoff component (surface runoff, subsurface lateral flow, and baseflow) are summed up to the total runoff, which is then the input to the routing model (Schulla, 2024).


## 4.4 Model calibration and evaluation

### 4.4.1 Calibration parameters

Here we summarize the parameters to calibrate in each module in *WaSiM*. In the snow module, the 11 key parameters that govern snow accumulation, snow ablation, wind-driven and gravitational redistribution of snow are calibrated (see Tab.3). In

the glacier module, the 7 key parameters that govern the initialization of glacier area and volume and the ice and firn melt are calibrated (see Tab.3). The average equilibrium line elevation (ELA, m a.s.l.) in the model is set to 2900 m a.s.l., according to the observations at the Langenferner glacier in 2006 (Autonome Province of Bolzalno, 2007). The volume-area scaling factor is set to 28.5, adopted from the work of Chen and Ohmura (1990), and the exponent is set to 1.375, according to the work of Bahr et al. (2015).


In the subsurface module, the bottom depth of the aquifer is calibrated to be 1.30 m, which must be identical to the bottom of the soil model in *WaSiM* (Schulla, 2024). The aquifer horizontal conductivity, storage coefficient, and colmation of the soil are homogeneous spatial grids and calibrated in the groundwater model. The properties for each soil layer and for each soil type are defined per grid cell, such as soil thickness, vertical hydraulic conductivity, and characteristics of the macropores. A

total of 13 parameters are calibrated for the entire module (Tab.4). Initial (manual) sensitivity analyses have shown that the remaining parameters are insensitive, and the default values suggested for each soil type from the user manual (Schulla, 2024)





are adopted for them.

All calibrated parameter values are given in Tables 3 and 4. It is worth mentioning that building up a physics-based, fully-distributed hydrological model requires numerous details about the catchment to constrain the model parameters, which is practically challenging. The purpose of our model is, however, not to reproduce exactly the catchment, but to understand the hydrological processes by using a reasonable parameter set, and that the simulated dynamics (relative changes) in the system should be reasonably consistent (Engdahl, 2024).

**Table 3.** Calibrated parameters of the snow and glacier modules in *WaSiM* (Schulla, 2024, 1997).

| Module | Parameter | Description | Value |
|---|---|---|---|
| Snow accumulation | TRS | Temperaure at which half of the precipitation falls as snow [°C] | 0 |
| Snow accumulation | Ttrans | Half of the temperature range from snow to rain [°C] | 2 |
| Snow ablation | LWINcorr | Correction factor for incoming long-wave radiation [-] | 1.2 |
| Snow ablation | LWOUTcorr | Correction factor for outgoing long-wave radiation [-] | 1.0 |
| Snow gravitational redistribution | i_lim | Max. deposition slope [°] | 55 |
| Snow gravitational redistribution | D_lim | Scaling for max.deposition [mm] | 2 |
| Snow gravitational redistribution | i_erosion | Min. slope for creating slides [°] | 50 |
| Snow gravitational redistribution | f_erosion | Fraction of snow pack forming the slide [-] | 0.007 |
| Wind-driven redistribution | Start azimuth | First quantile of wind direction [°] | 179 |
| Wind-driven redistribution | End azimuth | Third quantile of wind direction [°] | 217 |
| Wind-driven redistribution | cmin | Min. correction factor [-] | 0.5 |
| Glacier initialisation | VA_scale | Volume-area scaling factor [-] | 28.5 |
| Glacier initialisation | VA_exp | Volume-area exponential factor [-] | 1.375 |
| Glacier initialisation | WEchnge | Change rate of water equivalent per m [mm] | 1.8 |
| Glacier initialisation | ELA | Average equilibrium line elevation [m a.s.l.] | 2900 |
| Glacier melt | MF | Melt factor [mm] | 2 |
| Glacier melt | ice_min | Min. radiation coefficient for ice [mm Wh$^{-1}$ m$^2$ °C$^{-1}$ day$^{-1}$] | 0.0001 |
| Glacier melt | ice_max | Max. radiation coefficient for ice [mm Wh$^{-1}$ m$^2$ °C$^{-1}$ day$^{-1}$] | 0.0007 |





**Table 4.** Calibrated parameters of the unsaturated zone and groundwater model in *WaSiM* (Schulla, 2024, 1997).

| Module | Parameter | Description | Value |
|---|---|---|---|
| Unsaturated zone | kd | Storage coefficient of direct flow [-] | 2 |
| Unsaturated zone | ki | Storage coefficient of subsurface lateral flow [-] | 10 |
| Unsaturated zone | dr | Scaling factor applied to river drainage density [-] | 0 |
| Soil table | krec | Recession constant for the saturated conductivity with depth [-] | 1 |
| Soil table | PMThr | Macropore threshold [mm h$^{-1}$] | 2 |
| Soil table | MRedu | Capacity reduce [-] | 1 |
| Soil table | Soil thickness | Soil thickness [m] | 1.30 (5 layers in total) |
| Soil table | ksat | Vertical hydraulic conductivity [m s$^{-1}$] | $4\times10^{-4}$ to $6\times10^{-6}$ (forest area) $4\times10^{-4}$ to $5\times10^{-7}$ (grassland) $1.25\times10^{-5}$ to $2\times10^{-7}$ (rocks) |
| Groundwater | Input grid (.aq1) | Aquifer thickness [m] | 1.30 |
| Groundwater | Input grid (.kol) | Colmation of the soil [m s$^{-1}$] | $1\times10^{-6}$ |
| Groundwater | Input grid (.kx05) | Horizontal conductivity [m s$^{-1}$] | $5\times10^{-5}$ |
| Groundwater | Input grid (.ky05) | Horizontal conductivity [m s$^{-1}$] | $5\times10^{-5}$ |
| Groundwater | Input grid (.s03) | Storage coefficient [-] | 0.3 |

### 4.4.2 Calibration and evaluation procedure

The model is calibrated manually for each module in sequence (from surface to subsurface), similar to the procedure adopted for calibrating this type of model (fully-distributed, physically-based) in the work of Fatichi et al. (2015). Given the large number of parameters, the complicated surface-subsurface hydrological processes (snow-glacier-vegetation-river-unsaturated zone-groundwater), and the complex topography where the elevation ranges over 2000 m (1840 to 3760 m a.s.l.), no computationally intensive automatic calibration is used. We adopt a multi-signal calibration approach to calibrate each module in *WaSiM* with available relevant reference data, starting from top to bottom and from surface to subsurface processes. The manual calibration follows a parameter value perturbation approach: The parameters are first set to plausible default values and then successively perturbed until a satisfactory fit is obtained (see Section 4.4.3).

To calibrate manually each module in sequence, a total of ~250 simulations are performed. Each model simulation with a selected parameter set runs from 1 Oct 2006 to 15 Jan 2024 in a 25 m × 25 m gridded resolution at daily time steps for the whole catchment. The first 5 years (2006-2011) are taken as the warm-up period, (i) as the glacier module takes time to





be stabilized to produce reasonable simulations (Hofmeister et al., 2022), and (ii) due to the groundwater memory effect. The rest of the simulation (2012-2024) is used for model calibration and evaluation. Due to the long temporal and high spatial resolution, a complete simulation takes around 10 hours to run on a computer with 6 computational cores (Processor: 13th Gen Intel(R) Core(TM) i7-1370P 1.90 GHz).

### 4.4.3 Reference signals and performance criteria

In the multi-signal calibration, the simulated spatial SWE in *WaSiM* is calibrated to (i) the fractional snow-covered area derived from the MODIS daily snow cover maps (2002-2019) (Matiu et al., 2019) performed by Hofmeister et al. (2022) and (ii) the observed SWE at the in-situ stations from 2007 to 2024 (see Sections 3.1). The glacier module is calibrated to (i) the observed historic fractional glacier-covered area of the whole catchment (see Section 3.3), and (ii) the annual glacier mass balance of the Langenferner glacier (2007-2021) located in the Subcatchment S8 (Galos et al., 2017; Autonome Province of Bolzalno, 2007). The unsaturated zone and groundwater modules are calibrated to the observed groundwater hydrographs at the piezometers and to the river discharge. The discharge measured at the Upper Plima station (ID 450346), the station (ID 450499) before the extreme flooding event, and the official Plima station at the Zufallhut in Table 2 are adopted for evaluating the simulated discharge.

We select good model parameters via visual comparison between the model simulation and the observations for different reference data sets. We also evaluate the goodness-of-fit of the model in terms of reproducing the observed SWE, discharge, and groundwater heads with the Nash-Sutcliffe Efficiency (NSE) (Nash and Sutcliffe, 1970). Due to the uncertainties in the measured discharge and groundwater heads, the Spearman's rank correlation coefficient ($\rho$) is additionally adopted to evaluate the model performance, when the ranks in the observations and simulations are distinct (Gauthier, 2001).

## 5 Results and discussion

The final calibrated simulation covers the period from 1 October 2006 to 15 January 2024. Below, we discuss the observed and simulated results regarding the key aspects of surface-groundwater interactions.

### 5.1 Snow and glacier observations and simulations

In the snow module, the energy-balance method combined with the lateral wind-driven and gravitational redistribution algorithms (Schulla, 2024; Thornton et al., 2021; Shrestha et al., 2014; Warscher et al., 2013; Gruber, 2007) shows to outperform the one without redistribution effect, in terms of simulating the snow observations and the river discharge, especially during the high and low flow period (see Fig.A2 in the Appendix). The combined energy-balance method with snow redistribution algorithms is thus adopted for hydrological modeling. Figure 2 shows the observed and simulated snow and glacier changes and melt rates in the Martell Valley. We extract the simulated time series of snow water equivalent (SWE) from the spatial



gridded results at the locations of snow stations and compare them with the observations (see Section 3.1). The simulated SWE
captures well the observed snow dynamics at different altitudes (NSE: 0.87 for Zufritt Station, 0.74 for Zufallhuette Station;
Spearman's rank correlation coefficient: 0.93 for Zufritt Station, 0.88 for Zufallhuette Station). The simulation reproduces well
the interannual variability of snow accumulation and the severe snow droughts in the recent past, with e.g. simulated peak
snow accumulation in 2012, 2017, 2022, and 2023 being as low as 138, 103, 112, and 111 mm w.e., which are well below
the mean of 316 mm w.e. computed over the evaluation period 2011-2023 at the Zufritt Station (see Figs. 2ab). The snow
drought in 2022 was observed not only in the Martell Valley but also in the Po Basin in Italy (Avanzi et al., 2024). Hofmeister
et al. (2022) adopt the same snow parameter set and show comparable performance between the simulated snow cover and the
remote sensing data. The simulated snow dynamics and distribution are further used to simulate the hydrological changes in
this study.

Due to the lack of catchment-wide glacier mass balance observations, we use the observed fractional glacier coverage
(glacier area divided by the whole catchment area) as reference data for glacier module calibration (see Fig. 2c). The observed
fractional glacier coverage shows a declining trend since 1850, except the period around 1985 showing an increase. In con-
trast, the simulation first shows an increasing trend during the warm-up period and then decreases like the observations from
2014. The overestimated fractional glacier coverage at the model start is primarily because of some non-existent small glaciers
appearing due to preferential snow deposition along north-east exposed mountain ridges. Hence, the glacier model needs time
to stabilize and generate robust and reliable results. The simulated catchment glacier mass balance (zoomed-in plot in Fig.
2c) shows a typically strong seasonality and a significant decreasing trend of -126 mm w.e. year$^{-1}$ ($p$-value < 0.001, for the
evaluation period 2011-2023). As global warming continues, the glacier retreat is estimated to accelerate according to this
trend. Additionally, the glacier module is calibrated to the annual glacier mass balance of Langenferner in the Subcatchment
S8 (Fig.1). Apart from the warm-up years, the *WaSiM* glacier module captures the general dynamics of the glacier mass evolu-
tion in both mass growing (in 2014) and loss years. The model underestimates the annual glacier mass balance by around 8%
(annual mean simulation vs. observation: -935 vs. -1014 mm between 2011 and 2021 in Figure 2d).

The total annual liquid water input into the catchment (called "precipitation equivalent" or PEQ hereafter) is composed of
annual rainfall, snowmelt, and glacier melt. Figure 2e shows the annual magnitude of each input flux between 2007 and 2023.
We quantify the trend of each component starting from 2011 after the warm-up period. The annual rainfall, snow melt, and
glacier melt amount to 35%, 45%, and 20% of the simulated annual PEQ on average (2011-2023). The PEQ shows a decreas-
ing trend of -21 mm w.e. year$^{-1}$ ($p$-value = 0.13), indicating the decreasing water availability in this catchment over the past
decade. The annual trends of the components are: rainfall -4.5 mm year$^{-1}$ ($p$-value = 0.50), snowmelt -26 mm year$^{-1}$ ($p$-value
= 0.10), and glacier melt +10 mm year$^{-1}$ ($p$-value = 0.16). These results imply the increasingly important role of glacier melt
in sustaining the water availability in this and similar catchments, in particular in light of decreasing rainfall and snowmelt,
despite the trends not being yet statistically significant. With the ongoing global warming and accelerating glacier retreat, the
water availability in this catchment is likely to continue declining, posing water scarcity risks to the downstream ecosystems,



communities, and infrastructures (e.g. hydropower plants) (Avanzi et al., 2024).





**Figure 2.** Observed and simulated snow, glacier, and melt rates in the Martell Valley. (a-b) The observed and simulated daily snow water equivalent (SWE) in mm w.e. at the (a) Zufritt and (b) Zufallhuette Stations (elevations: 1851 and 2265 m a.s.l. in Table 2). (c) The observed and simulated annual fractional glacier coverage (i.e. glacier area divided by the whole catchment area) for 1850-2019 and 2007-2023. The zoomed-in plot shows the simulated catchment daily glacier mass balance (glmb). (d) The observed annual glacier mass balance of Langenferner compared with the simulation of subcatchment S8. (e) The simulated catchment annual total input flux (snowmelt, glacier melt, and rainfall) between 2007 and 2023. Warm-up period: 2007-2010, evaluation period: starting from 2011.





## 5.2 Hydrological response of river and shallow groundwater

Figure 3a shows the observed daily river and groundwater levels at the measuring stations in a glaciated upper subcatchment (S8 in Fig.1, 31% of glacier coverage in this subcatchment) and their responses to the simulated total PEQ between Nov 2022 and Nov 2023. The groundwater boreholes IDs 4479 and 4478 are located at a similar elevation (2455 m and 2454 m a.s.l.)
and are both around 30 m away from the river on the same side (see Fig.1 and Tab.2). During the winter period between Nov 2022 and Apr 2023, the river shows a slow recession and a low water level, which corresponds to $<0.5$ m$^3$ s$^{-1}$ of streamflow discharge or to $<3.5$ mm day$^{-1}$ (given the subcatchment area of 12.3 km$^2$). This low winter streamflow is classically observed for many high-elevation catchments ($>2200$ m a.s.l.) (Schmid, 2024). It can result from groundwater or be sustained by other hydrological processes, such as leaked glacier melt or ground heat-induced snowmelt (Müller et al., 2024). The groundwater
level at borehole ID 4479 shows rather stable and slow dynamics during the cold months. In contrast, the neighboring borehole ID 4478, which is 115 m away from borehole ID 4479, shows a strong recession and drop-down of the heads. These results underline the highly heterogeneous subsurface properties and associated groundwater dynamics in high alpine catchments.

In May 2023, the river water level and both groundwater heads (IDs 4479 and 4478) were responsive to the early snowmelt
and rainfall. Again, the groundwater borehole ID 4478 shows a much higher water level fluctuation (around 0.8 m) than the river gauge and the other borehole ID 4479. However, starting from glacier melt onset in June and in particular after glacier melt intensification in July (see Fig.3b), the two boreholes show similar dynamics and responses to the total PEQ, in contrast to their different behaviors in the winter season. During the warm period, the river and the groundwater heads are all responsive to the snow and glacier melts and the high rainfall events, such as those in August and September. Overall, the shallow
groundwater level can increase as fast as the river level during the peak melts and high rainfall events in this glaciated site. The results imply that a precise partitioning of the snow and glacier melt and the rainfall between the surface and subsurface is required to achieve reliable hydrological simulations.





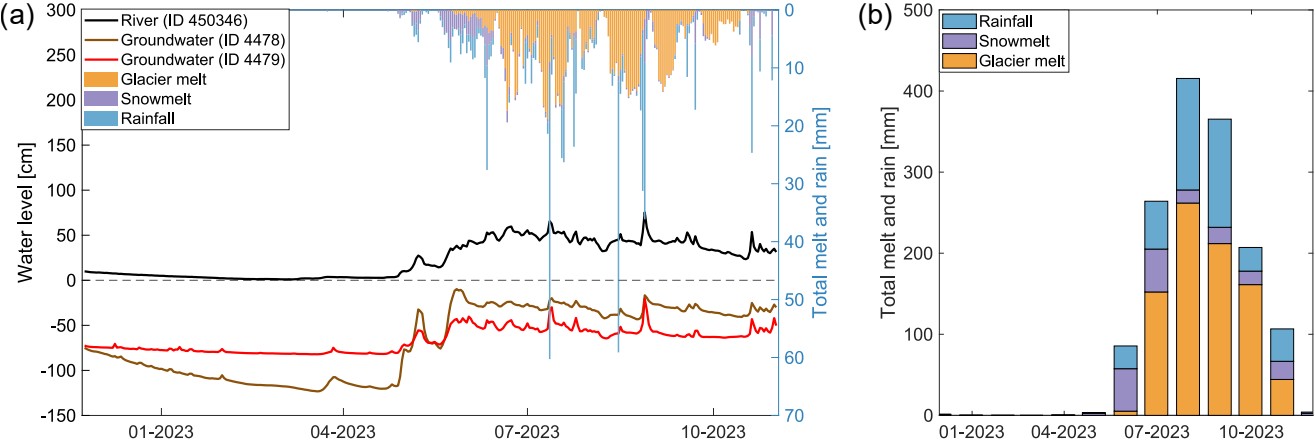

**Figure 3.** Hydrological response of the observed river water level and groundwater level to the simulated total rainfall and meltwater from snow and glaciers between Nov 2022 and Nov 2023 in the glaciated Subcatchment S8 (Fig.1) in the Martell Valley. (a) The observed daily river and groundwater level hydrographs and the simulated daily meltwater and rainfall in this subcatchment. The black line denotes the river water level measured at the gauge Upper Plima (ID 450346) in cm day$^{-1}$. The red and brown lines denote the groundwater level measured at the bore ID 4479 and ID 4478 in cm day$^{-1}$. The stacked bars on the top denote the simulated daily glacier melt, snowmelt, and rainfall in mm day$^{-1}$. (b) The simulated monthly total snowmelt, glacier melt, and rainfall in this glaciated subcatchment in mm month$^{-1}$. The yellow blocks denote the glacier melt, the purple blocks denote the snowmelt, and the blue blocks denote the rainfall.

## 5.3 Groundwater observation and simulation

Figure 4 shows the observed and simulated groundwater levels of the five installed groundwater boreholes in the catchment (see Fig.1 and Tab.2), including simulations with or without subsurface lateral flow in the unsaturated zone (also called "interflow"). All observed boreholes measure the heads of the unconfined shallow aquifers that have a depth to water table (DTW) within 1.2 m. Results show that the best simulations (see Fig.4) at all sites are achieved when the subsurface lateral flow is forced to 0 in the model. The Spearman's rank correlation coefficients for the best simulations against the observed hydrographs of each

bore are 0.89, 0.84, 0.41, 0.61, 0.81 for boreholes IDs 4479, 4478, 4476, 4473, and 4672 (Figs.4a-e). The simulations captured well the observed temporal dynamics, such as the head response to the early and peak snowmelt and to the high rainfall events. The magnitudes of water table fluctuations are also well modeled at most of the sites, except for borehole ID 4478 where the strong recession behavior in the winter period is challenging to simulate (Fig.4b).

When the subsurface lateral flow is allowed in the model, the simulated groundwater hydrographs show a delayed response to the early and peak snowmelt, underestimation of the observed groundwater level, and struggle to reproduce the observed temporal dynamics (see Fig.4). Extensive parameter testing shows that this delayed simulated response cannot be attenuated by choosing other parameters for the subsurface. By calibrating the hydrological model to the observed groundwater levels, these experimental results provide new insights that the subsurface lateral flow in the unsaturated zone plays a minor role in





contributing to the streamflow in the study site. This could be because of the presence of shallow groundwater within a depth of 1.2 m: as a consequence, the subsurface lateral flow is most likely minor in such shallow thickness. More importantly, these results challenge the commonly adopted hydrological modeling approach, i.e. the role of soil in streamflow simulations may have been overrated by giving high attention to the surface runoff and subsurface lateral flow, but using a simplified representation of baseflow (Gao et al., 2023).

**Figure 4.** Observed and simulated groundwater levels at the five stations in the catchment, simulated with and without subsurface lateral flow in the unsaturated zone. The groundwater heads are recorded as depth to water table (DWT) below the land surface. The black lines denote the observed hydrographs; the red lines denote the simulations without subsurface lateral flow (by setting the parameters of *dr*=0 and *krec*=1 in the *WaSiM* model); the blue and purple dashed lines denote the simulations with subsurface lateral flow (by slightly perturbing the *dr* and *krec* parameters).



Given that the groundwater characteristics are spatially homogeneous (i.e. horizontal hydraulic conductivity, storage coefficient, aquifer thickness) and that the neighboring groundwater boreholes (e.g. IDs 4479 and 4478) have the same land use and soil properties in the model, the simulated different groundwater dynamics at both sites can be assumed to be dominated by the surface properties, namely topography. To further investigate the reasons why the groundwater recession cannot be simulated at some locations of the model domain, we compute the Topographic Wetness Index (TWI, Eq.1), which is a key measure for understanding the topographic control on the catchment hydrological processes (Pourali et al., 2016; Beven and Kirkby, 1979). The TWI of the whole catchment ranges from 2.7 to 23.4, with an average of 6.0. The higher the TWI is, the wetter the modeling cell tends to be. We quantify the relationships between the TWI and the simulated annual mean groundwater level of the five observed boreholes and their neighboring cells. All sites show a positive relationship between these two variables (see Fig.A4 in the Appendix).

Figures 5ab show the TWI spatial distribution in the part of the catchment where the groundwater boreholes are located. In Figure 5a, borehole ID 4479 has a TWI of 7.1 and its simulation shows an adequate performance in capturing the observed groundwater level variability. In contrast, borehole ID 4478 is located in a flood plain and has a TWI of 10.7, indicating that this area is more likely to store water in the model. This may explain why the model struggles to release water at this location in the subsurface in the winter season (see Fig.4b). We acknowledge that there might be other hydrological processes occurring that reduce subsurface storage, such as by preferential flow paths, but these are not present in the model. Additionally, the computed TWI value is subject to the resolution and uncertainty of the DEM data, which are used for calculating the local slope, flow direction, and flow accumulation. Given that (i) the DEM data and the model have a spatial resolution of 25 m, which could be large for such complex topography, (ii) the DEM data were available in 2006 while the groundwater heads were measured in 2022/2023, and (iii) the DEM data uncertainty could be larger than 1 m, which is the measured elevation difference between the two boreholes (IDs 4479 and 4478, see Tab.2), the DEM data uncertainty may hence impact the modeled hydraulic gradients between the two sites.

Besides the model and DEM uncertainty, a key question is how reliable and representative the observed data at specific locations are for calibrating a comparable distributed model. In our study, the hydrograph of one borehole (ID 4479) can be simulated by the grid cell corresponding to the exact borehole coordinates (Figs.5ac); for borehole ID 4478, which shows a nearly full subsurface and very high TWI value in the model (Figs.5ad), the model simulations cannot reproduce the observed dynamics. It is hence reasonable to check the simulations not only for the grid cell containing the point observation but also those of neighboring cells (see Figs.5cd), to account for uncertainties related to DEM resolution, and the evolution of the river networks.





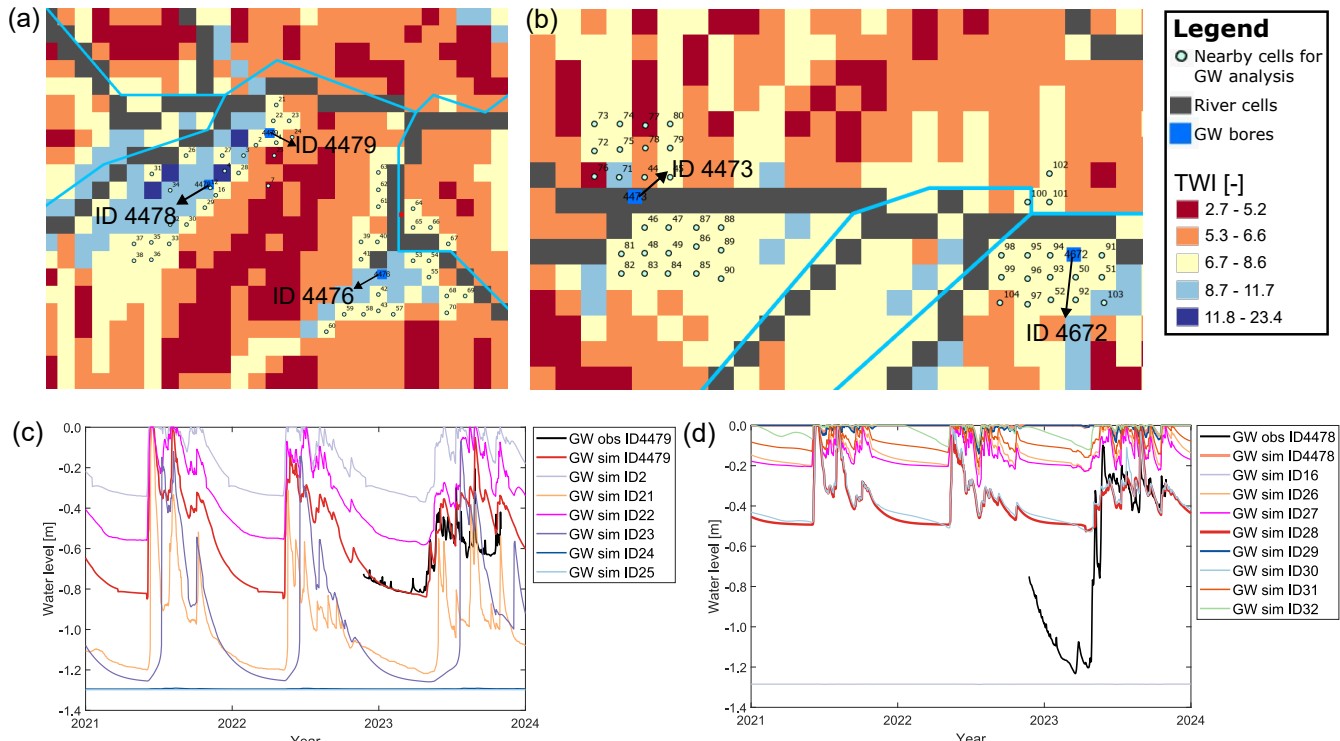

**Figure 5.** Spatial distribution of the Topographic Wetness Index (TWI) in part of the catchment where the groundwater boreholes are located, and their observed and simulated groundwater hydrographs. (a-b) The zoomed-in TWI maps show the five observed boreholes: (a) IDs 4479, 4478, 4476, and (b) IDs 4473 and 4672. The spatial resolution of the modeled grid cells is 25 m × 25 m. The blue lines represent the observed river networks; the black cells represent the computed river networks with *Tanalys* (Schulla, 2024), which is an add-on *WaSiM* software. The little dots denote the locations where the groundwater level simulations are analyzed, and the numbers next to them are their assigned IDs. (c-d) The observed and simulated groundwater hydrographs between 2021 and 2023 for borehole IDs 4479 and 4478 at their observed locations and their nearby locations. The black lines denote the observed hydrographs; the red lines denote the best simulations among all nearby locations; the other color lines denote the simulations besides the best simulations.

## 5.4 River observation and simulation

Figure 6 shows the observed and simulated river discharge of three subcatchments S3, S7, and S8 (in Fig.1). The subcatch-
ment S3 is close to the end of the catchment where the Plima River outlet is located. Despite the interrupted observations, the simulation (at Station ID 450499 in S3) captures well the observed discharge dynamics as shown in Figure 6a and Figure A3a, with the Spearman's rank correlation coefficient of 0.79; the relatively low NSE of 0.50 can be explained by the occasional overestimation of the peak flow and underestimation of the discharge magnitude in 2020. We cannot further comment on winter model performance for this station due to little reference data.




The official discharge station of Plima is located in the subcatchment S7. Also at this location, the model captures the observed temporal dynamics of the discharge well, with a Spearman's rank correlation coefficient of 0.90 and NSE of 0.63 (Fig.6b and Fig.A3b). Again, the model tends to underestimate the observed discharge and to occasionally overestimate the high flow rates. This could partly be because the observed river discharge is computed from river water levels with a rating

curve (Hofmeister et al., 2023b) where high flows are typically underrepresented (Di Baldassarre and Montanari, 2009). The occasional overestimation of the high flow could result from forcing the subsurface lateral flow to 0, and hence lead to overestimation of surface runoff for high-rainfall events. In winter, the simulations show very low values (mean discharge between December and March from 2018 to 2024: 0.59 $m^3$ $s^{-1}$ or 2.0 mm day$^{-1}$ given the subcatchments area of 25.3 $km^2$), which (see above) is not uncommon for corresponding elevations. No winter flow was observed before 2018, as the Plima Station was not

in operation in winter (Fig.6b). The observed low winter flow since 2019 could result from (i) the river water level being too low to be measured by the water sensor at the gauging station or (ii) the sensor being frozen under low temperatures.

The simulated and observed discharge at the river gauge Upper Plima (ID 450346) in the glaciated upper subcatchment S8 has a Spearman's rank correlation coefficient of 0.90 and NSE of 0.76 (Fig.6c and Fig.A3c). The observed winter low

discharge at this station was discussed in Figure 3 in Section 5.2. Modeling the river discharge at this glaciated subcatchment is challenging. The model underestimates the very low discharge in winter and overestimates the high discharge in the warm period sometimes. However, the model captures the temporal dynamics of the observed discharge in terms of response timing to the early and peak snowmelt and high rainfall events.



**Figure 6.** Observed and simulated river discharge of (a) subcatchment S3 close to the end of the catchment, where the river Plima's outlet is located, (b) subcatchment S7 where the official discharge station of Plima is located, and (c) the glaciated upper subcatchment S8. The subcatchment locations are shown in Figure 1. The black lines denote the observed river discharge at each station. The red lines denote the simulated discharge at the outlet of each subcatchment.

Figure 7 shows the simulated surface runoff ("direct flow" in the model), groundwater baseflow, and the proportion of baseflow in the total streamflow (i.e. baseflow divided by the sum of direct flow and baseflow. Note that the subsurface lateral flow is forced to 0 after calibration to the groundwater heads) in the subcatchments S2 and S3. These subcatchments are located





close to the Plima River outlet. Shown are also the subcatchments S7 and S8, which are the upper headwater subcatchments (in Fig.1). The glacier coverages of the 4 subcatchments (S2, S3, S7, S8) are 11%, 0, 32%, and 31% (glacier area/subcatchment

area). Both the direct flow and baseflow show interannual and seasonal variations. The direct flow plays a dominant role in the magnitude of the river discharge in all subcatchments (Figs.7a-d). The estimated baseflow shows a very low magnitude compared with the direct flow, but a clear seasonality and responds to the peak snowmelt and high rainfall events (Figs.7e-h). Besides that, the baseflow in the subcatchment without glaciers (S3) shows a higher variability than for the subcatchments with glaciers (S2, S7, and S8). It is worth noting that the groundwater boreholes, which are used to calibrate the unsaturated zone

and groundwater model in *WaSiM*, are mainly located in the upper and the middle of the catchment (S7, S8), which is rather flat and assumed to have groundwater storage. As the subsurface properties are parameterized as homogeneous, the simulated baseflow in the lower catchment (e.g. S2, S3) may contain higher uncertainty than those in the upper catchment (e.g. S7 and S8). Starting from the peak snowmelt, the simulated baseflow increases but the magnitude is still relatively low. This could be because of the relatively small hydraulic gradient between the groundwater table and river water table, given the shallow river

depth of <10 cm and the narrow river width of <2 m (observed in the field) in the glaciated headwater subcatchment (e.g. S8).

Despite the low baseflow, the proportion of baseflow in the total streamflow emphasizes the important contribution of baseflow in the streamflow, especially in the winter low-flow period (Figs.7i-l). The groundwater baseflow in winter provides up to 30-40% of streamflow in the subcatchments S7 and S8 (i.e. upper headwater subcatchments), up to 10% of streamflow in the

subcatchment S2 and >40% of streamflow in the subcatchment S3 (where the main river outlet is located).







**Figure 7.** Simulated surface runoff (direct flow), baseflow, and the proportion of baseflow in the streamflow in the subcatchments S2 and S3, which are close to the Plima River outlet, and the upper headwater subcatchments S7 and S8. (a-d) direct flow in mm day$^{-1}$, (e-h) baseflow in mm day$^{-1}$, (i-l) the proportion of baseflow [%] in the total runoff (i.e. baseflow divided by the sum of direct flow and baseflow. The subsurface lateral flow is forced to 0 after calibration to the groundwater heads). The locations of the subcatchments are shown in Figure 1.

## 5.5 Spatial analysis of key hydrological changes

Figure 8 shows the simulated key hydrological spatial changes of monthly percolation, groundwater recharge, groundwater level, groundwater exfiltration to the river, and river infiltration into the groundwater for February (lowest flow), June (peak snowmelt), September, and December (low flow) in 2023, which is the year when the groundwater levels were measured. The percolation represents the vertical free drainage into the groundwater. The groundwater recharge is calculated for each grid cell as the water balance between the vertical and horizontal inflow and outflow. The percolation and groundwater recharge are close to 0 across the catchment in February during the lowest flow period. The groundwater level is within 1 m depth at the locations close to the river networks and the Zufritt lake in February, but it is lower than the simulation depth limit (i.e.





1.3 m) for other locations. During the peak snowmelt period in May/June, the percolation reaches the highest value in the year and the groundwater gets recharged at most of the locations except at very high elevations and in steep areas; the groundwater level increases accordingly, especially near the river channels and in the bottom of the valley. The percolation, recharge, and groundwater levels remain at a high level until September 2023, when the total liquid input (rainfall, snowmelt, glacier melt) into the catchment is still high (see Fig.3b). After that, they gradually turn to a low state in December 2023 (Fig. 8).

The spatial results also show the seasonal changes in the interactions between groundwater and river channels (see Fig.8). The groundwater exfiltrates to the river all year round for the main river channels, and the exfiltration rate diminishes with increasing elevation. The groundwater recharges the river channels even at the highest elevations in June, i.e. during peak snowmelt, when the active stream network is fully developed. However, in February, groundwater only recharges the main river channels, and the exfiltration shows disconnectivity in the river networks at high elevations (part of the stream network is not active). In contrast to the spatial pattern of groundwater exfiltration to the river, river infiltration into the groundwater only occurs at the main river channel at the bottom of the valley. This could be because baseflow (i.e. groundwater exfiltration to the river) can occur in *WaSiM* at any stream network grid cells with adjacent groundwater cells. However, river infiltration into groundwater can only occur at groundwater cells which are next to main river channels. This follows from the assumption in the model that the flow gradient is usually directed from groundwater to rivers in the headwater subcatchments; this gradient could only be reversed if there are large amounts of external inflows to the routing channels (Schulla, 2024). Overall, these results show that the physics-based fully-distributed model is powerful in visualizing and analyzing the spatial patterns of hydrological changes in great detail (such as where and when groundwater recharge occurs) and the seasonal changes of spatial interactions between groundwater and river.





**Figure 8.** Spatial distribution of the simulated key hydrological changes of monthly percolation, groundwater recharge, groundwater level (in depth to the water table), groundwater exfiltration to the river, and river infiltration into the groundwater in the Martell Valley in February, June, September, and December 2023.




## 5.6 Challenges and opportunities for modeling high-alpine glaciated environment

The physics-based fully-distributed model enables a detailed analysis of hydrological changes in the surface and subsurface and thus contributes to new insights into hydrological process understanding. Such detailed hydrological modeling is especially necessary for high alpine glaciated catchments. As the water towers, these landscapes are highly sensitive to the impact of climate change (Marzeion et al., 2018; Bliss et al., 2014). Plausible projections of future water availability in the glaciated environment are crucial for guiding water resources management and mitigating adverse climate change impacts.

However, building a physics-based, fully-distributed model requires numerous details and field data (e.g. vegetation, soil properties, glacier, river networks, fine-scale DEMs, fine spatial coverage of climate and hydrological stations) for constraining and calibrating the model parameters. This task can be particularly challenging for high alpine glaciated catchments, where the hydroclimatic records are often short and interrupted, the topography is highly complex and can change significantly even at fine spatial resolution, the surface and subsurface properties are spatially heterogeneous (Ramskogler et al., 2023), and the boundary conditions are often unknown.

### 5.6.1 Model calibration

The study is among the first few physics-based fully-distributed hydrological modeling for a comparable high alpine glaciated catchment, with a multi-signal calibration approach and particularly focusing on calibrating the subsurface with the groundwater level observations. The manual calibration is chosen here because automatic calibration is computationally prohibitive and the number of parameters to calibrate would not align with the amount of available reference data for such an automatic calibration. Given the manual calibration and corresponding parameter uncertainty, the retained calibration corresponds to a physically reasonable parameterization but not to any sort of optimal parameterization. As stated earlier, the purpose of our model is to understand the range of variability and the possible interplay of hydrological processes by using a reasonable or possible parameter set. The detailed comparison with observed data (SWE, annual glacier mass balance, river discharge, groundwater heads), the quantified relative contribution of baseflow in streamflow, and the spatial and temporal interactions between surface water and groundwater show that the simulated system dynamics (relative changes) are reasonably consistent (Engdahl, 2024).

### 5.6.2 New insights into alpine hydrological processes

The observed shallow groundwater level hydrographs show a very quick increase after input events; the observed variations are as quick as those observed for river water levels. This observation can be thought of as what is called runoff generation via groundwater feedback (Ala-aho et al., 2017; Appels et al., 2017). Such groundwater feedback is rarely explicitly parameterized in models, which implies that there is a need to improve the parametrization of the subsurface in many commonly used





hydrological models. In fact, a simple linear representation of the subsurface, as used in many commonly adopted simplified (conceptual) hydrological models, cannot reproduce such a behavior (Horton et al., 2022). A vertically-resolved representa-

tion (e.g. via Richards equation as here) of subsurface flow cannot easily overcome this problem; rather we would need new parameterizations, for example, by including fast preferential flow pathways (Sivelle et al., 2025; Mazzilli et al., 2019) or piston-flow-like behavior (Wang et al., 2018; Paniconi and Putti, 2015), or approaches that explicitly account for saturation-area effects (such as in the dynamic Topmodel approaches as implement in the DECIPHeR model (Coxon et al., 2019)).

Despite the observational and modeling challenges, we demonstrated how such a model can contribute to new insights into hydrological processes at high elevations. For example, we find that the subsurface lateral flow in the unsaturated zone plays a minor role in streamflow generation at our study site. This phenomenon is revealed for the first time by introducing shallow groundwater level observations into the model calibration. Adopting a multi-signal calibration approach, especially by including the groundwater level observations in the calibration, thus contributes to new insights into the hydrological process

understanding between the surface and subsurface. This result may challenge the commonly adopted hydrological modeling approach, which may have often overrated the role of soil in partitioning the water flow. In other words, existing studies, by emphasizing the components of surface runoff and subsurface lateral flow, might have underestimated the role of baseflow (Gao et al., 2023). This finding may depend on the catchment size and subsurface characteristics (e.g. presence of shallow groundwater). It needs to be further tested, in particular for larger catchments of areas in the order of hundreds of km$^2$, where

subsurface lateral flow may not be negligible (Schulla, 2024). Besides that, the Martell Valley is a relatively dry catchment (with a mean annual precipitation of around 1000 mm) among the European Alps. The role of groundwater could be different for the catchments with a much higher annual precipitation (e.g. more than 2000 mm per year) as compared to a relatively dry environment. Given the lithology of Martell Valley dominated by crystalline bedrocks and shallow soil formation (Ramskogler et al., 2023; Berra et al., 2012), the findings here contribute to understanding the hydrological processes in the high alpine

catchments with similar climatic and geologic settings.

### 5.6.3   Recommendations on groundwater piezometer installation and evaluation

By calibrating the subsurface module of the model to the observed groundwater level, we find that the model can be well calibrated to fit the observed data at some sites, while for others, the observations are hard to reproduce. For example, the sim-

ulations at and around borehole ID 4478 struggle to release water from the subsurface storage during the winter period. One explanation might reside in the fact that, given the assumed homogeneous subsurface properties, the simulated groundwater level dynamics completely depend on the surface properties, i.e. the topography. To gain further insights and to quantitatively understand the wetness potential of the modeled grid cells, we computed the TWI. The results show that the TWI value at and around the borehole ID 4478 location is much higher than the catchment average (10.7 vs. 6.0); accordingly, the model

struggles to empty the subsurface at this location. To avoid similar pitfalls for future high-elevation groundwater modeling with





observed data, we suggest the following:

(i) It is recommended to first compute the TWI values of the catchment (at anticipated modeling resolution) based on the available DEM data and then to decide the locations where to install the groundwater piezometers (in addition to any additional field information). This helps avoid particularly wet and dry locations for which the model would struggle to simulate ground- water level changes. Due to the high resolution of spatial input data (needed to build complex models in complex terrain), the model has a low tolerance for observation errors, such as the borehole coordinates and the DEM data, which are key for generating the spatial raster inputs for the model (e.g. river networks, local slope, flow direction, and accumulation). Given the high challenge of correcting the DEM data at fine spatial resolution, installing the piezometers at locations that are promising for comparison with the model, i.e. by adapting the modeling task to the DEM data uncertainty, could potentially help improve the reliability of groundwater simulations.

(ii) Grid-cell-scale simulated groundwater levels might reliably represent larger-scale average groundwater dynamics despite not being in agreement with observed groundwater levels. This is similar to snow modeling, where simulated snow heights may not be comparable to in-situ observations at the exact station location (but still represent adequate performance for the whole catchment (Dong, 2018)). Due to the intrinsic uncertainty in the observed data and model structure and parameters, it is recom- mended to calibrate the observed groundwater hydrographs not only to the simulation at the exact location but also to consider neighboring cells. In this study, the horizontal distances between the river and the installed boreholes (IDs 4478, 4479) are ∼30 m for detecting surface water-groundwater interactions, while the model resolution is 25 m. This only allows for one grid-cell distance, which is challenging to simulate in the model (e.g. the observed borehole may fall in a river cell like borehole ID 4473 in Figure 5). This modeling spatial resolution is chosen by balancing the trade-off between the computational demand and the representation of the site characteristics. Due to the DEM uncertainty, the generated river network grid cells (derived from DEM data) may not match the exact river locations in reality. The distance between the river and boreholes in the model may hence differ from reality even though the exact observed borehole coordinates are adopted in the model. Therefore, the groundwater simulations of the neighboring cells near the borehole locations should be considered in groundwater model calibration as well.

### 5.6.4 Interactions between groundwater and surface water

The observed river station at Upper Plima (ID 450346) in the headwater subcatchment, shows a very low water level in the winter season, which is $<0.5$ m$^3$ s$^{-1}$ (equivalent to $<3.5$ mm day$^{-1}$, see Fig.6b). This very low discharge could be due to groundwater baseflow or other hydrological processes, such as leaked glacier melt or ground-heat-induced snowmelt (Müller et al., 2024). Practically, it is challenging to measure very low winter discharges, which could be affected by high observational uncertainty: the river stage level is measured with a pressure sensor, which may stop working when the stream runs very low or in freezing temperatures.





The simulated baseflow shows a low magnitude of 0.2-0.6 mm day$^{-1}$ at the study site (subcatchments S2, S7, S8 in Fig.7), but a clear interannual cycle and seasonality. The very low baseflow could be because the groundwater piezometers are mainly installed in the upper headwater subcatchments. The river channels are relatively shallow and narrow (<10 cm depth and <2 m width) in this terrain, thus leading to low groundwater exfiltration to the river. As the subsurface properties are modeled as homogeneous and mainly calibrated with the groundwater observations in the upper subcatchments, the simulated baseflow

in the lower subcatchments may contain higher uncertainty. Nevertheless, the spatial results show the seasonal changes in the interactions between groundwater and river (see Fig.8). The river channels are recharged by groundwater during the high flow months (e.g. June and September 2023), even for river branches at very high elevations. However, the groundwater recharge diminishes with increasing elevation in the winter time and mainly recharges the main river channel at the bottom of the valley (see Fig.8). This is in line with the typically observed retraction of the active stream network during winter months at compa-

rable elevations (Michel et al., 2022; Paillex et al., 2020). Despite the underestimated magnitude of the groundwater baseflow, these modeling results contribute to understanding the temporal and spatial variations of the interactions between groundwater and surface water.

## 6   Conclusions

In this study, we propose methods to simulate the complex surface-subsurface hydrological processes and to quantify the interactions between the key hydrological components (snowmelt, glacier melt, river discharge, groundwater flow) in a high-elevation glaciated catchment (Martell Valley). We adopt the physics-based, fully-distributed hydrological model, *WaSiM*, with an integrated groundwater module. Despite the observational and computational challenges of building such a detailed model, this study contributes to new insights into hydrological processes in high alpine environments and makes suggestions for future

hydro(geo)logical modeling.

       Results show that the shallow alpine groundwater level responds nearly as quickly as streamflow to snowmelt and to heavy rainfall events (Fig.3). As this quick groundwater response is rarely simulated by the model, this highlights the need for improved subsurface parametrization in hydrological models. By incorporating the shallow groundwater level observations into

the multi-signal calibration, surprisingly, the subsurface lateral flow in the unsaturated zone is found to play a minor role in streamflow generation (Fig.4). Integrating groundwater observations in the multi-signal calibration thus contributes to new insights into the hydrological processes in such an environment. Additionally, our results underline the challenges of integrating point-scale groundwater observations into the calibration of a fully-distributed hydrological model. To overcome this challenge, we suggest, in particular, to compute relevant topographic metrics (relevant for flow transfer in the model) prior to

choosing any field site for piezometer installation, in addition to field information (Fig. 5). This will enable to avoid installing piezometers at particularly wet and dry cells in the model and to adapt hydrological modeling tasks with the DEM data uncertainty. Besides integrating model-based consideration into field site location, we also recommend comparing any groundwater



level simulations to a range of neighboring model cells, adjacent to the exact piezometer location; this is a simple yet efficient method to address the intrinsic observational and modeling uncertainties.


Despite the underestimated magnitude of groundwater baseflow, the physics-based fully-distributed model emphasizes the crucial contribution of baseflow to winter streamflow and provides valuable insights into the temporal and spatial variations of the interactions between groundwater and surface water. Overall, by overcoming the modeling challenges and integrating point-scale groundwater observations in model calibration and evaluation, this study sheds light on surface-subsurface hydrological
processes in high-elevation glaciated environments and highlights the importance of improving subsurface representation in hydrological modeling.

**Appendix A**

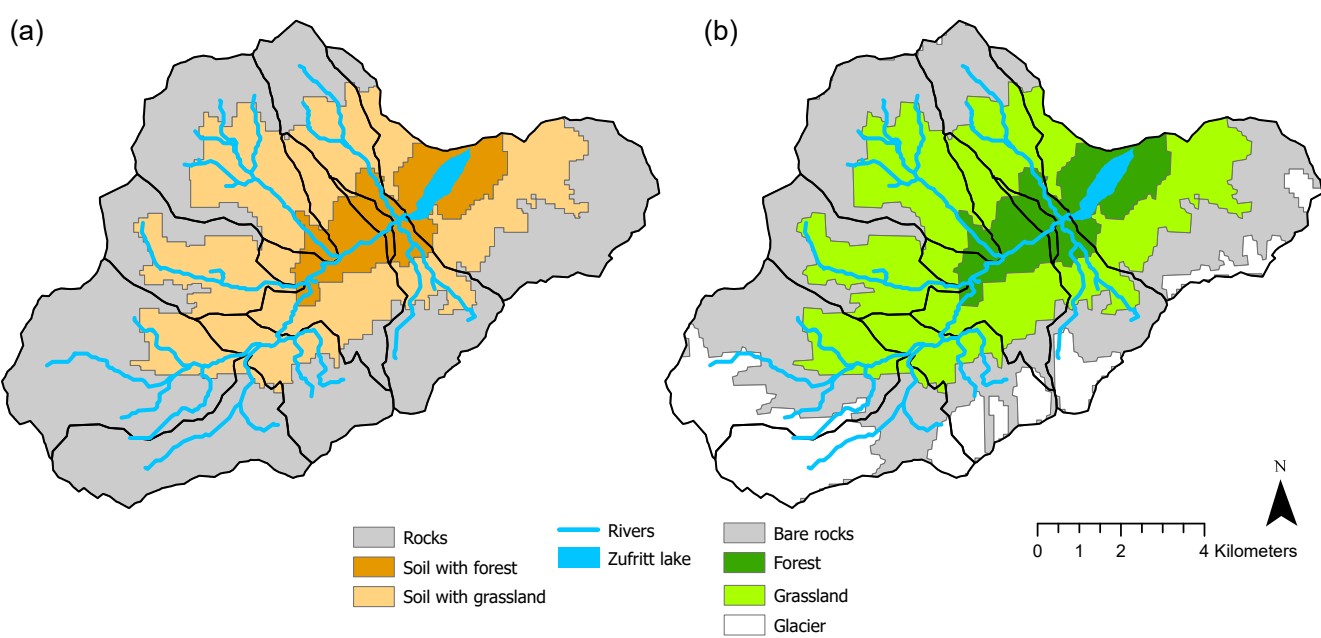

**Figure A1.** The maps of (a) soil types that are classified as rocks, soil with grassland formation, and soil with forest, and (b) land use and land cover that are classified as bare rocks, glaciers, grassland, and forest in 2018 in the Martell Valley.


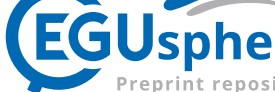

**Figure A2.** (a-b) The observed and simulated daily snow accumulation by using the energy-balance method with and without snow redistribution algorithms activated in *WaSiM*. (c-e) The observed and simulated daily discharge with and without snow redistribution algorithms activated for the subcatchments (c) S3, (d) S7, and (e) S8 (see Fig.1). The model performances of the simulations are summarised in Table A1. The black dot and line denote the observation, the red line denote the simulation with snow redistribution, and the blue line denote the simulation without snow redistribution.



**Table A1.** The model performance of the simulated snow accumulation and discharge with (Y) and without (N) snow redistribution algorithms activated in *WaSiM* (Schulla, 2024, 1997).

| Station | NSE (Y) | NSE (N) | Spearman's correlation coefficient (Y) | Spearman's correlation coefficient (N) |
|---|---|---|---|---|
| Zufritt (snow in Fig.A2a) | 0.87 | 0.86 | 0.94 | 0.93 |
| Zufallhuette (snow in Fig.A2b) | 0.76 | 0.74 | 0.88 | 0.88 |
| ID 450499 (discharge in Fig.A2c) | 0.50 | 0.35 | 0.87 | 0.84 |
| Plima (discharge in Fig.A2d) | 0.63 | 0.60 | 0.90 | 0.89 |
| ID 450346 (discharge in Fig.A2e) | 0.76 | 0.69 | 0.90 | 0.87 |

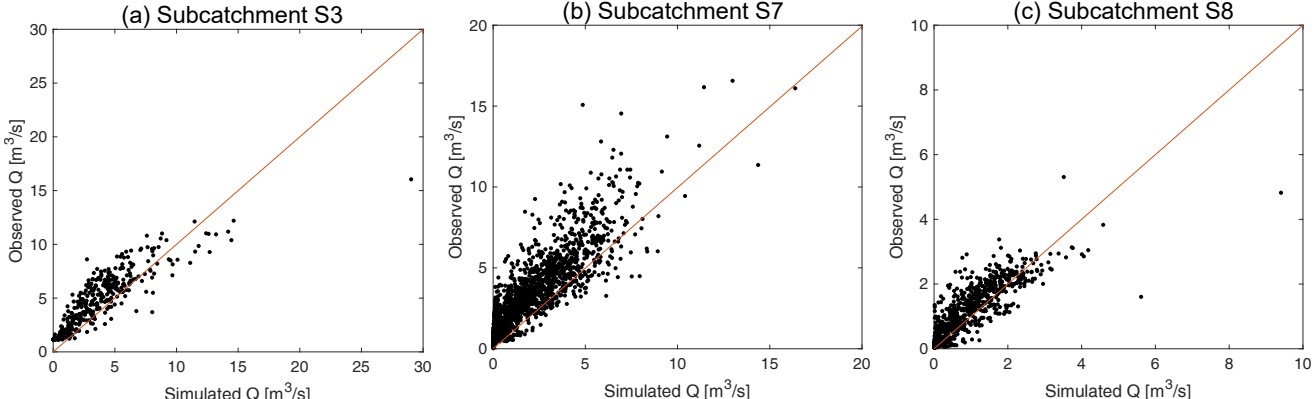

**Figure A3.** Scatter plots of the observed against simulated discharge (Q) in the subcatchment S3, S7, and S8, corresponding to Figure 6.



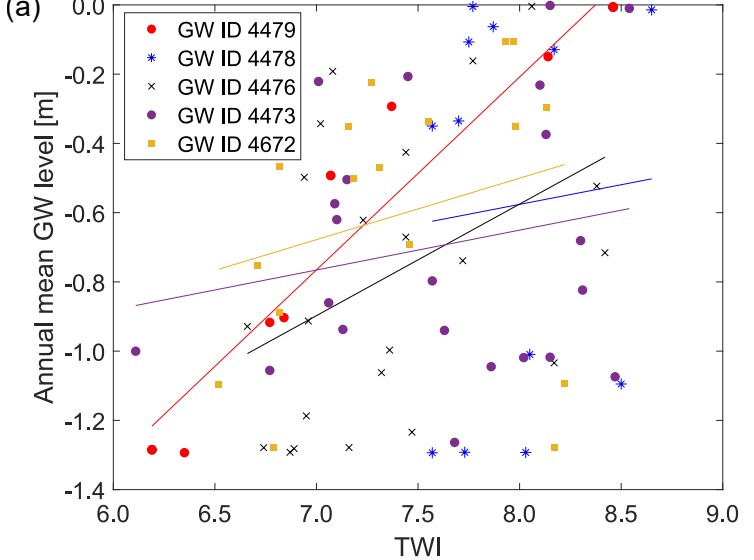

| | slope | $R^2$ | *p*-value |
|---|---|---|---|
| GW ID4479 | 0.54 | 0.94 | 0.01 |
| GW ID4478 | 0.18 | 0.02 | 0.69 |
| GW ID4476 | 0.37 | 0.25 | 0.02 |
| GW ID4473 | 0.10 | 0.03 | 0.43 |
| GW ID4672 | 0.17 | 0.06 | 0.33 |

**Figure A4.** (a) The relationship between the Topographic Wetness Index (TWI) and the simulated annual mean groundwater (GW) level between 2011 and 2023 of the five observed boreholes and their neighboring cells in the model. Each color represents the data of each borehole. The lines denote the trends quantified for each site with the data at the observed location and their neighboring cells. The slope, the linearity, and the significance level of the trends are shown in (b).

*Code and data availability.* A compiled version of the hydrological model *WaSiM* is available at http://www.wasim.ch/en/ (no source code
available). The ΔSnow model (v1.0.2) used for transforming the snow height data to snow water equivalent is accessible in the R package
"*nixmass*" (Winkler et al., 2021). The MATLAB function *mannkendall* (v1.0.0) is adopted for quantifying the temporal trends of the hydro-
logical variables (Collaud Coen and Vogt, 2021; Collaud Coen et al., 2020). The groundwater level data and river discharge data measured
during the project will be made available in an appropriate data archive upon acceptance of the paper. The discharge data of the official Station
Plima are obtained from the Autonomous Province of Bozen (Bolzano) - South Tyrol. The observed climate data at Station Langenferner are
obtained from the Institute of Atmospheric and Cryospheric Sciences of the University of Innsbruck in Austria. The data at Station Careser
are obtained from Meteo Trentino in Italy (http://storico.meteotrentino.it/web.htm?ppbm=T0065&rs&1&df). Other station data are obtained
from the Autonomous Province of Bozen (Bolzano) - South Tyrol in Italy. The Digital Elevation Model (DEM) data in 2006 are obtained
from Province of Bolzano (di Bolzano Alto Adige, 2006). The land use and land cover data of the site in 2018 are obtained from the Corine
Land Cover Data Center (CORINE Land Cover, 2018). The soil profile and attributes in 2018 are extracted from the Harmonized World
Soil Database (version 2.0) (FAO & IIASA, 2023; Ramskogler et al., 2023). The geological information of the site is obtained from the map
produced by Berra et al. (2012). The glacier coverage maps of the site are obtained from Knoflach et al. (2021).

*Author contributions.* All authors participated in the conceptualization, methodology, data acquisition, design of the experiment, and discus-
sion on the results. XF carried out the formal analyses and investigations. The hydrological *WaSiM* model is calibrated by XF and FH. XF



wrote the original manuscript and visualized all the results, all co-authors contributed to the editing and review. BS and GC supervised the work and acquired the project funding.

*Competing interests.* The authors declare that there is no competing interest in this work.

*Acknowledgements.* The authors acknowledge the valuable discussions about the hydrological modeling results with the *WaSiM* developer Jörg Schulla. The authors thank the technician Mr. Michael Tarantik and the students at the Technical University of Munich for measuring and obtaining the hydrological data in the Martell Valley. The authors thank the project partners in Eurac Italy for providing the information on the catchment land cover, geology, and soil properties, also thank the official authorities of Autonomous Province of Bozen (Bolzano) - South Tyrol, Institute of Atmospheric and Cryospheric Sciences of the University of Innsbruck, Meteo Trentino in Italy for providing the research data. XF, FH, and GC acknowledge the support of the German Research Foundation (DFG) research unit (FOR2793/2) investigating the "Sensitivity of High Alpine Geosystems to Climate Change since 1850" (SEHAG) under grant CH981/3-2. XF and BS acknowledge the Swiss National Science Foundation (SNSF) grant (200020E-204030) for "SEHAG Subproject 2 -Impact of climate change on groundwater storage in high Alpine catchments: from observation to model predictions".



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
