# Peer review of "Physics-based simulation of hydrological processes in a high-elevation glaciated environment focusing on groundwater"

_EGUsphere, 2025_

## Author Comment (AC1)

Comment #1: The reliance on manual calibration for a complex, fully-distributed model with numerous parameters is a significant limitation. While the authors justify this choice due to computational constraints and data availability, the manuscript would benefit from a more thorough discussion of the implications of manual calibration on parameter uncertainty and model robustness. For instance, how sensitive are the key findings (e.g., minor role of subsurface lateral flow) to parameter choices? A sensitivity analysis, even if limited, could strengthen the credibility of the results.

Comment #2: The multi-signal calibration approach is a strength, but the sequential calibration process (surface to subsurface) may introduce biases. The authors should discuss potential dependencies between modules (e.g., snow module influencing groundwater recharge) and how these were addressed to ensure consistency across the calibration steps.

Answer to #1 and #2: Thanks for the suggestion. We calibrated the physics-based hydrologic model manually by following a top-down approach and module by module. We first performed manual sensitivity analysis on the key parameters in each module, and then focused on calibrating the sensitive parameters in detail. The insensitive parameters are assigned with the default values suggested in the model user manual or adopted from the literature. We will add the results of the suggested sensitivity analysis on the key parameters and strengthen the discussion of the adopted approach in the revised version.

The sequential calibration of module by module is a commonly adopted logical procedure to calibrate such a fully-distributed physics-based hydrological model. The module by module calibration offers a good diagnostic power, as it isolates which modules (e.g., snowmelt, glacier melt) are causing discrepancies between observation and simulation. This allows an incremental validation, as each module can be tested and validated before integrating with the next. By doing so, errors in specific processes (e.g., snow or glacier melt) can be addressed without compromising other well-performing modules. Through the simplification of the calibration strategy, the parameter interactions are reduced, which leads to more stable model results.

Despite the calibration is sequential, we rerun the whole model (including all modules) each time when a parameter in a module is perturbed, and we focus on the model performance to the observed variables of that module. For example, when we calibrate a parameter in the snow module, we run the whole hydrological model and all temporal and spatial hydroclimatic outputs are produced, but we focus on the model performance compared to the snow water equivalent at the observed stations and spatial snow coverage. In this way, the hydrological processes between the modules are interconnected and the consistency is ensured. We will improve the articulation of the model calibration in the revised manuscript.

Comment #3: The finding that subsurface lateral flow plays a minor role in streamflow generation is intriguing but requires further scrutiny. The assumption of homogeneous subsurface properties (e.g., hydraulic conductivity, storage coefficient) across the catchment may oversimplify the complex geology of the Martell Valley, which is noted to be heterogeneous (Section 2). This assumption could bias the model toward underestimating lateral flow. The authors should explore whether spatially variable subsurface parameters, informed by available geological data, could alter this conclusion.

Answer: The subsurface of our model is not fully homogeneous. We applied essential heterogeneity in the subsurface by calibrating the hydraulic conductivity in each soil layer and for each soil type. More detailed heterogeneity of the subsurface characteristics in finer spatial

resolution is challenging to apply, as this information is vastly unavailable in the high-elevation catchments with glaciers. Many studies on the topic of stochastic hydrogeology show the challenges of estimating the spatially variable subsurface parameters even within an extremely small experimental area. Assigning heterogeneous subsurface properties horizontally and vertically in the ungauged areas involves numerous assumptions. We therefore applied a conservative approach to ensure the essential subsurface heterogeneity. Exploring the impact of spatially variable subsurface parameters is out of scope of this study.

Comment #4: The model's inability to reproduce the strong winter recession at borehole ID 4478 (Section 5.3) suggests limitations in capturing preferential flow paths or other subsurface processes. The manuscript would benefit from a deeper discussion of alternative mechanisms (e.g., macropore flow, fractured bedrock) that could explain this discrepancy, potentially supported by literature or additional field observations.

Answer: We will strengthen the discussion on these likely alternative mechanisms in the revised manuscript.

Comment #5: The challenges of integrating point-scale groundwater observations into a distributed model are well-articulated, but the proposed solutions (e.g., using TWI to guide piezometer placement, comparing neighboring cells) need more rigorous evaluation. For instance, how representative are the TWI-based recommendations for other high-alpine catchments with different topographic or geologic characteristics? A sensitivity analysis of TWI resolution or comparison with other topographic indices could enhance the generalizability of these recommendations.

Answer: We suggest calculating TWI as an additional information to support the decision on where to install the piezometers, besides the expert knowledge and field information. The TWI is mainly derived based on the Digital Elevation Model data and does not relate to the geological data. The spatial resolution of the TWI (or the hydrological model resolution) should be determined by the individual site characteristics such as topography. In our study, we tested the spatial resolution of 25x25m, 50x50m, and 100x100m, and we finally adopted 25x25m by balancing the trade-off between the computational intensity and the site characteristics. Future studies are welcome to test this approach and adopt a reasonable spatial resolution based on their site condition and research aims.

Comment #6: The manuscript highlights the mismatch between observed and modeled river networks due to DEM uncertainties (Section 5.6.3). This issue could significantly affect groundwater-surface water interactions, yet it is only briefly addressed. A quantitative assessment of DEM uncertainty (e.g., comparing simulations with different DEM resolutions) would strengthen the discussion and provide more concrete guidance for future studies.

Answer: We did test different spatial resolutions of 25x25m, 50x50m, and 100x100m in the hydrological model beforehand and finally adopted the 25x25m, by balancing the trade-off between the computational resources and the details of the topography, which is reasonably captured by this DEM resolution. More detailed DEM leads to significantly higher computational demand, which is not plausible and unlikely to significantly alter the conclusions.

Comment #7: The authors note that the Martell Valley is relatively dry compared to other Alpine catchments (Section 5.6.2), which may limit the applicability of findings to wetter environments. Similarly, the lithology (crystalline bedrocks, shallow soils) may not be representative of other high-alpine settings. The discussion should more explicitly address the conditions under which

the key findings (e.g., minor role of lateral flow, rapid groundwater response) are likely to hold, potentially by comparing with studies in contrasting catchments.

Answer: We will strengthen the discussion and articulation of the generalisation of the research findings in terms of climatic and hydrogeologic conditions.

Comment #8: The manuscript claims that the rapid groundwater response is rarely simulated by hydrological models (Section 6), but this statement requires more substantiation. A brief review of other physics-based models (e.g., HydroGeoSphere, ParFlow) and their ability to capture such dynamics would contextualize the novelty of WaSiM's performance and clarify the need for improved subsurface parameterization.

Answer: We agree with this point and will strengthen the discussion by comparing this study with similar modeling efforts in the high alpine studies.

Comment #9: The underestimation of winter baseflow (Sections 5.4, 6) is attributed to shallow river channels and homogeneous subsurface parameterization, but observational uncertainties in low-flow measurements (e.g., sensor limitations in freezing conditions) are also significant (Section 5.6.4). The manuscript should more clearly disentangle model limitations from observational uncertainties, possibly by discussing the reliability of winter discharge data or exploring alternative data sources (e.g., tracer studies) to validate baseflow contributions.

Comment #10: The claim that baseflow contributes significantly to winter streamflow (up to 40% in some subcatchments, Section 5.4) is compelling but relies on model simulations rather than direct observations. Additional evidence, such as isotopic or chemical tracers, could corroborate this finding and enhance confidence in the model's representation of baseflow dynamics.

Answer to #9 and #10 on additional evidence: We will strengthen the discussion regarding the winter baseflow from shallow groundwater by considering the suggested points. Our water stable isotope data of groundwater and river sampled in winter low-flow condition (2022/23, 2023/24) do show consistent values and thus provide another evidence that shallow groundwater provides winter baseflow. We, however, did not show these data in the manuscript as they were not sampled regularly in high frequency. We agree that this aspect will be better articulated in the revised version.

Comment #11: The introduction is comprehensive but lengthy, with some repetition (e.g., challenges of alpine hydrology are mentioned multiple times). Streamlining the introduction to focus on key gaps and the study's objectives would improve readability.

Answer: We will improve the readability of the introduction by considering the given suggestion.

Comment #12: Section 5.6 is titled "Challenges and opportunities for modeling high-alpine glaciated environment," but it primarily discusses challenges. Explicitly addressing opportunities (e.g., leveraging remote sensing, integrating machine learning for parameter estimation) would balance the narrative and highlight future research directions.

Answer: Thanks for the suggestion. We will address this aspect by enhancing the discussion on the future opportunities, such as the newly emerged approach on integrating machine learning technique into physics-based modelling.

Comment #13: The use of abbreviations (e.g., PEQ, TWI, DTW) is frequent, and a glossary or table defining these terms would aid readers unfamiliar with the terminology.

Answer: Thanks for the suggestion, but there are actually only a few abbreviations in this study, and we feel that a glossary table might be unnecessary in this case.

Comment #14: Figure 8 is visually rich but overwhelming due to the number of panels. Consider splitting it into two figures (e.g., one for percolation/recharge, another for groundwater level/exfiltration/infiltration) or using a subset of months to improve clarity.

Answer: We acknowledge this suggestion but would prefer to keep them in a panel to enable to intercompare the results between the months and the inter-related variables.

Comment #15: Table 3 and Table 4 list calibrated parameters but lack units for some parameters (e.g., "Scaling for max.deposition" in Table 3). Ensuring consistency in units and providing brief explanations for less intuitive parameters would enhance accessibility.

Answer: The unit of "Scaling for max.deposition" in Table 3 is already given right after this term; it is in mm. We ensured to provide all units for all parameters in the tables. For the ones that do not have a unit, we marked them as [-]. A brief explanation is provided after each parameter as well by following the model user manual to allow readers for cross-checking them in detail in the manual.

Comment #16: The caption for Figure 5 could clarify that panels (c-d) show simulations for multiple grid cells, as this is not immediately obvious from the figure alone.

Answer: We will revise this information in the caption for improving readability.

Comment #17: I highly recommend to discuss the SWAT-MODFLOW papers which integrates SW-GW and cite below paper:

Assimilation of sentinel-based leaf area index for modeling surface-ground water interactions in irrigation districts

Answer: Thanks for the suggested literature and we will consider it. However, the suggested literature focuses on coupled modeling in the irrigation fields, while our study focuses on the high alpine hydrology in the glaciated environment. Despite the topic of modeling, the research approach, focus, and site conditions are largely different.

---

## Author Comment (AC2)

We thank the reviewer for their time in reviewing this manuscript. We carefully considered the comments, which point out some of the key challenges of modelling groundwater in high-elevation alpine catchments. We will provide point-by-point answers during the revision of the manuscript (should a revision be encouraged by the editor).

Hereafter, we provide a detailed discussion of the main comments of the reviewer:

**(1) Details of high-elevation aquifers and subsurface heterogeneity:**

The majority of the comments ask for a detailed/improved representation in the model of various types of aquifers (hillslope aquifers, stream-wide aquifers, comment on Line 230), of varying depths of different types of aquifers and their boundaries (comments on Lines 225, 256, 362 and 433), and of spatial subsurface heterogeneity distributed in the complex terrains. The reviewer asks for modelling the deep aquifers (e.g. several tens of or 100 m depth), which might exist in the Quaternary sediments in part of the catchment.

We would like to emphasize that we focus on modelling the shallow groundwater in this study. The requested details of the deep aquifers by the reviewer are not represented in our model (despite the fact that the model is potentially able to represent some of these details for simulation), because this detailed information is vastly not available. In fact, this information is even rarely available in the temperate climate and rather flat catchments, and not only in catchments with high elevation (>2000 m a.s.l.), with complex topography, and glacier coverage, such as the Martell Valley.

Not including the deep aquifers in the model does not mean losing the essence of the model at the headwater catchment scale. Shallow aquifers play important roles in sustaining winter streamflow, despite the relatively low magnitude in winter (Fig. 13 in Müller et al. 2022 for a proglacial outwash plain similar to Martell Valley). Modeling shallow groundwater with a hydrological model enables new understanding of the near subsurface hydrological processes and of how water partitioning is shaped by spatial and temporal surface water-groundwater interactions.

We include the essential heterogeneity in the subsurface by calibrating the hydraulic conductivity for each soil layer and for different soil types. Obtaining effective (calibrated) hydraulic conductivities is a prerequisite for a hydrologic model to reliably simulate observed streamflow, and this is a key challenge in physics-based models that often rely on pedotransfer functions to estimate hydraulic conductivity (Paschalis et al. 2022).

Heterogeneities occurring at a smaller scale (e.g. 25 m) are very difficult to characterize, as widely demonstrated by the vast literature in stochastic hydrogeology, starting from the detailed investigation of the famous sites of Borden and Columbus, for example (Rehfeldt et al. 1992; Woodbury, et al. 1991).

As pointed out by van Tiel et al. (2024), it is challenging to describe processes controlling cryosphere-groundwater interactions in alpine catchments due to a lack of observed data and modelling challenges. As a matter of fact, bucket models have been applied to the Martell Valley so far, in which the physically based description of groundwater fluxes is even more limited in comparison to our work (Schaffhauser et al, 2024; Puspitarini et al, 2020).

**(2) Remote sensing data:**

Remote sensing data (suggested by the reviewer) could potentially be used for estimating groundwater changes; we could think of GRACE data, but the low spatial resolution (>100km) is too coarse for high-elevation catchments characterized by a catchment area smaller than 100 km$^2$. The coarse resolution of the available remote sensing products for groundwater changes leads to high uncertainty, and it is hard to be adopted in a fully distributed hydrological model. Another limiting aspect of satellite-based gravimetry is the superimposition of gravity changes in glaciated high mountains. Here, changes in mass are caused by glacier melt, seasonal snow cover, and fluctuations in subsurface storage (Voigt et al. 2021).

**(3) Physics-based modeling:**

We respectfully disagree with the comment that our model is not physics-based. Part of the difference in perspective might stem from what we mean by "physics-based": We mean a model in which we describe groundwater movement according to the groundwater flow equation, while water movement in the unsaturated zone is described by Richard's equation. This approach is not fully 3-D as done by other software (e.g., HydroGeoSphere (Thornton et al, 2022)), but it still represents a step forward in comparison to bucket models, and it is, in our view, a good compromise between data availability and model complexity. This will be clearly specified in the revised version.

We acknowledge that WaSiM has an unusual specific requirement that the soil thickness must be kept the same as the aquifer bottom in the model, as the lower soil layers are taken as the saturated zone. Despite this numerical constraint, WaSiM shows to be useful to model the shallow aquifer (Fig.4 in this study), and it even outperforms the external coupling of surface (WaSiM) and subsurface (HydroGeoSphere) models for an alpine catchment, which show significant challenges in modelling the observed shallow groundwater heads within 2-5 m depth (see Fig. 4 in Thornton et al. 2022, 2021).

With the available information and data, we show in this study how to build this kind of fully distributed physics-based hydrological model for both surface and shallow subsurface processes in a high-elevation glaciated catchment, which is a missing piece in the literature.

**(4) Adoption of a hydrological model (WaSiM) instead of a hydrogeological model:**

In this manuscript, our research objective is to use a state-of-the-art hydrological model for groundwater modelling in such a high-elevation glaciated alpine catchment, with all its limits. We will make it clearer that this is not a groundwater model study. We adopt the hydrological model WaSiM, rather than a hydrogeological model (e.g. HydroGeoSphere, HGS), due to numerical and hydrological process understanding reasons.

The most relevant work on this topic among the handful modeling studies is the work of Thornton et al. (2022) who externally coupled the WaSiM (surface model) with HGS (subsurface model) for a high alpine catchment. That is, the snowmelt and rainfall generated from WaSiM are given as inputs to HGS. They show the challenges to neither get a good match of groundwater observations nor the streamflow (Figs. 3 and 4).

Firstly, the HGS is highly computationally intensive, challenging to calibrate, solves Richards equations in 3-D, and needs a good quality of geological model and soil depth map, where this information is hardly available and simple assumptions have to be applied for high alpine studies. Most importantly, they pointed out that "Whilst the soil zone is very small compared to the unconsolidated and consolidated geological formations volumetrically, it likely exerts a

disproportionately strong hydrological influence via its influence on the partitioning of liquid water at the surface" (p.7 in Thornton et al. 2022). This is why we adopt a hydrological model over a hydrogeological model; we aim to get good partitioning of liquid water. Interestingly, by switching off the subsurface lateral flow in the hydrological model, we find good match for both groundwater and streamflow observations. Therefore, we need the hydrological model for such process understanding.

Compared to existing studies (Schaffhauser et al, 2024; Thornton et al, 2022, 2021; Puspitarini et al, 2020), the further added value is that (i) we build an integrated hydrological model with WaSiM, which avoids the uncertainty introduced by external coupling between a surface model and a subsurface model, (ii) we model the surface and shallow subsurface hydrological processes in a high-elevation alpine catchment, with glacier coverage, which enables to quantify the hydrological impact of glacier melt, and (iii) we model the subsurface (unsaturated and saturated zones) in a physics-based approach, given that most hydrological models represent the subsurface in a conceptual way.

**(5) Baseflow and subsurface lateral flow:**

Regarding the rather low baseflow mentioned by the reviewer, our groundwater piezometers are mainly installed in the upper headwater subcatchments, where the groundwater baseflow is potentially lower than the locations close to the catchment outlets and the lower part of the catchment (see the conceptual diagram in Fig. 1 in van Tiel, et al. 2023). In fact, we observe in the snow accumulation period a decrease in the groundwater table, which is relatively well captured by the model for ID4479, while the model underestimates the drop in the GW level for ID4478. This shows that the shallow aquifer described in the model may locally overestimate the storage. However, the shallow piezometers do not fall dry, and the experimental evidence does not exclude a connection between the shallow aquifer and the river during the snow accumulation period. Therefore, we cannot conclude, as pointed out by the reviewer, that the baseflow is uniquely generated in the winter months by a deeper circulation path. Additionally, the water stable isotope data collected in winters 22/23 and 23/24 for both the groundwater and the river display consistent values. Such results are not shown in the manuscript because they do not have a high temporal resolution, but they support the hypothesis of a connection between the shallow aquifer and the river during the winter period. We will circulate this point more clearly in the manuscript.

Regarding the subsurface lateral flow, it may not be negligible in the hillslope processes mentioned by the reviewer. In our study, the piezometers are installed in a rather flat area, where the subsurface lateral flow is found to play a minor role by calibrating the model to the observed groundwater heads. This finding could relate to the site characteristics and the observed locations. We will further clarify and discuss these points in the manuscript.

Despite all the limits, this research is among the first on performing such a detailed physics-based modelling study for both surface and shallow subsurface hydrological processes. We agree to better articulate and tone down some of the statements as pointed out in the revised manuscript.

**References**

Müller, T., Lane, S. N., & Schaefli, B. (2022). Towards a hydrogeomorphological understanding of proglacial catchments: an assessment of groundwater storage and release in an Alpine catchment. Hydrology and Earth System Sciences, 26(23), 6029-6054.

Paschalis, A., Bonetti, S., Guo, Y., & Fatichi, S. (2022). On the uncertainty induced by pedotransfer functions in terrestrial biosphere modeling. Water Resources Research, 58(9), e2021WR031871.

Puspitarini, H. D., François, B., Zaramella, M., Brown, C., & Borga, M. (2020). The impact of glacier shrinkage on energy production from hydropower-solar complementarity in alpine river basins. Science of The Total Environment, 719, 137488.

Rehfeldt, K. R., Boggs, J. M., & Gelhar, L. W. (1992). Field study of dispersion in a heterogeneous aquifer: 3. Geostatistical analysis of hydraulic conductivity. Water Resources Research, 28(12), 3309-3324.

Schaffhauser, T., Tuo, Y., Hofmeister, F., Chiogna, G., Huang, J., Merk, F., & Disse, M. (2024). SWAT‑GL: A new glacier routine for the hydrological model SWAT. JAWRA Journal of the American Water Resources Association, 60(3), 755-766.

Thornton, J. M., Therrien, R., Mariéthoz, G., Linde, N., & Brunner, P. (2022). Simulating fully‑integrated hydrological dynamics in complex alpine headwaters: potential and challenges. Water Resources Research, 58(4), e2020WR029390.

Thornton, J. M., Brauchli, T., Mariethoz, G., & Brunner, P. (2021). Efficient multi-objective calibration and uncertainty analysis of distributed snow simulations in rugged alpine terrain. Journal of Hydrology, 598, 126241.

van Tiel, M., Aubry-Wake, C., Somers, L., Andermann, C., Avanzi, F., Baraer, M., ... & Yapiyev, V. (2024). Cryosphere–groundwater connectivity is a missing link in the mountain water cycle. Nature Water, 2(7), 624-637.

Voigt, C., Schulz, K., Koch, F., Wetzel, K. F., Timmen, L., Rehm, T., ... & Flechtner, F. (2021). Introduction of a superconducting gravimeter as novel hydrological sensor for the Alpine research catchment Zugspitze. Hydrology and Earth System Sciences, 25(9), 5047-5064.

Woodbury, A. D., & Sudicky, E. A. (1991). The geostatistical characteristics of the Borden aquifer. Water Resources Research, 27(4), 533-546.

---

## Author Comment (AC3)

**General Comments**

The manuscript by Fan et al. presents a detailed and technically sophisticated application of a fully distributed, physics-based hydrological model, the Water Balance Simulation Model (WaSiM) to investigate surface–subsurface hydrological interactions in the glacierized catchment of Martell Valley, South Tyrol, European Alps. This study addresses a significant gap in our understanding of groundwater dynamics in alpine cryospheric environments. I appreciate the authors' effort in undertaking this important and challenging task.

The strength of this manuscript is the use of extensive observational data, often lacking in high-mountain environments. The comprehensive implementation of WaSiM to simulate both surface and subsurface hydrological processes is impressive. However, the manuscript would benefit from a clear articulation of its novel scientific contributions in relation to existing studies. I suggest including a dedicated comparison with similar modelling efforts to better highlight what is gained or potentially lost by the chosen approach.

Additionally, the limitations of applying such a model in complex mountain terrain, particularly with respect to spatial resolution, assumptions regarding subsurface properties, and data coarseness, should be discussed in detail with a separate section addressing model uncertainties, assumptions (e.g. uniform aquifer thickness). Further elaboration on key methodological decisions, such as the use of dual melt approaches, justification for subsurface parameterization, and the model's ability to represent delayed responses in groundwater will also help strengthen the manuscript.

Answer: We thank the reviewer for the constructive and reasonable feedback on our manuscript, and their appreciation in taking on this challenging task of performing a fully-distributed physics-based surface-subsurface modeling of the high-elevation glaciated environment.

We fully agree that a clear articulation of our study's novel scientific contribution in relation to the existing studies, and a dedicated comparison with the similar modeling efforts, would be greatly beneficial for highlighting the strengths and limitations of the chosen approach. We also agree to add in-depth discussions on the aspects mentioned in the comments, such as modeling assumptions regarding the subsurface properties, justified choices on spatial resolution, subsurface parameterization, and key methodological decisions in the revised manuscript.

**Specific Comments**

Comment #1: The authors position their study as one of the first to develop and implement a physics-based modelling framework to simulate surface–subsurface interactions in a glacierized environment. I agree with this claim, as there are indeed only a limited number of comparable studies. However, I recommend that the authors include a dedicated section comparing their approach with similar studies to clarify what is gained (or lost) by using this approach.

Answer: We agree that a dedicated section on comparing our study with the existing handful of similar modeling efforts will be beneficial to strengthen the manuscript. We will add this subsection in the revision.

Comment #2: The authors provide information about the catchment surface conditions, e.g. 40% bare rock, 34% grassland, and so on. I recommend that the authors include similar information about the valley floor settings (sediment-filled region), as most of the movement occurs in this

region. Additionally, providing information on the land cover (bare rock, grassland, forest) in the map (Figure 1) can help readers better understand the area and relevance for hydrology.

Answers: We agree to provide further information on the valley floor settings into the manuscript, and add the land cover information into the Figure 1.

Comment #3: In Section 4.1 and Table 3: the description of the snow/rain partitioning scheme using the parameters *TRS* (i.e. temperature at which half of the precipitation falls as snow [0 °C]) and *Ttrans* (i.e. half of the temperature range from snow to rain [+2 °C]) could benefit from further clarification or simplification. Some readers who are less familiar with hydrological modelling may find the current phrasing difficult to understand. I suggest explicitly stating that this defines a linear transition of snow fraction from 100% at –2 °C, 50% at 0 °C, and 0% at +2 °C. Additionally, the authors may explain the use of two different approaches (EB and T-index) for snow and ice melt, even though both processes are forms of cryospheric melt that are influenced by similar energy exchanges?

Answer: Thanks for the helpful suggestion. We will explicitly explain these two parameters by following the suggestion.

Ideally, the energy balance method should be adopted for both snow and ice melt processes. However, the energy balance method (with snow redistribution) is only available for snow melt in the model, so we adopt the extended temperature-based approach which enables to include the radiation information for simulating the ice melt. We agree to further elaborate on these method decisions in the revision.

Comment #4: 4.2. Unsaturated zone and groundwater model: In a mountain environment, especially when glacierized, the use of uniform aquifer properties ignores the highly heterogeneous nature of the subsurface system. Such assumptions may hold true in plains, where natural depositions are relatively uniform, but not in mountainous environments. The authors later in the results (L361–362, and on several other occasions) imply the heterogeneity of the subsurface properties. Also, why are vertical flows modelled but not horizontal flows in the unsaturated zone? Please explain. Are the authors sure about the absence of permafrost in the upper parts of the study area?

Answer: The mountainous environments can be highly heterogeneous as shown in the different groundwater heads responses in Figure 3. However, the detailed spatial subsurface properties are vastly unknown. Therefore, we could only include essential spatial heterogeneity in the model, i.e. by calibrating the hydraulic properties of different soil layers and soil types. Such compromised assumptions are adopted due to rare data available in the subsurface spatially. We agree to further discuss such assumptions regarding subsurface properties, data scarcity, and their limitations in the revision.

Here we adopt the Richard's equation to solve water flow in the unsaturated zone, which is 1-D vertical flow in the soil columns. In fact, most of the existing hydrological models (based on our knowledge) only simulate the 1-D vertical flow in the unsaturated zone but no horizontal flow. This could be due to the complexity of solving 3-D flow in the soil and the vertical flow is the primary process to recharge groundwater. The subsurface lateral flow is generated in a conceptual way in the model depending on the soil water content, local slope, and hydraulic conductivity.

In the highest parts of the Martell Valley, permafrost causes thawing and freezing in the underground. However, this is not included in our model due to lack of spatially distributed soil

temperature data. We will further elaborate and discuss this model assumption and its related uncertainties in the hydrological simulations in the manuscript.

Comment #5: 4.3. Streamflow generation: The model computes surface runoff, subsurface lateral flow, and baseflow separately and routes them using a reservoir cascade. This lumped treatment may oversimplify the interactions and timing differences among flow components.

Answer: The surface runoff and the subsurface lateral flow are routed to the streamflow using a reservoir cascade approach with a flow travel time concept applied (Schulla, 2024). The baseflow is generated and transmitted through the adjacent groundwater and river cells in a physics-based approach. We will further clarify these model specifics and its uncertainty in the manuscript.

Comment #6: Since groundwater/subsurface water dynamics are central to the study, I suggest the authors justify the use of 1.3 m as subsurface thickness across the entire catchment. The region is dominated by bare rock, grasslands, forest, and glaciers. It is obvious that subsurface conditions and thickness are not uniform. I believe the thickness of 1.3 m is low for such regions, especially in valley floors, with former glacial depositions or landslides. The authors mention numerical reasons and limitations, I understand that. In that case, the authors need to do a sensitivity test to see the influence of variable thickness on subsurface flow and storage, and on the overall outcomes. Additionally, the use of 1.3 m contradicts Figure 3(a), as the hydraulic head of borewell ID 4478 is around the same depth.

Answer: The adopted uniform soil thickness across the catchment is a simplified assumption, which is calibrated based on the groundwater levels observed at the five groundwater piezometers. It is challenging to assign spatially varying subsurface depths in the ungauged area, e.g. the valley floors. We will further discuss this assumption, the subsurface properties, and data coarseness in detail in the manuscript.

We agree to add the result of the sensitivity test of the soil thickness to justify our calibrated results, and its influence on the subsurface flow, storage, and the overall outcomes.

For Figure 3(a), this is not a contradiction though. The lowest groundwater head is about 1.2 m while the adopted soil thickness (subsurface thickness) is 1.3 m. This is due to the specific setup required by the WaSiM software that the soil thickness must be the same as the aquifer depth, as the bottom part of the soil layers represent the saturated zone.

Comment #7: The model ran at 25×25 m resolution. Isn't the soil profile from the global database (Harmonized World Soil Database), which has a resolution of 1 km, too coarse, particularly in a topographically complex region? The use of such data in a 77 km$^2$ area may not represent local settings (like moraine or similar glacial deposits). Please clarify.

Answer: The adopted soil data are the ones that are available for Martell Valley, despite the rather coarse spatial resolution of 1km. It is challenging to get more detailed soil profile data for this area. We adopt a high spatial resolution of 25 x 25 m for the modeling task to consider the detailed topography impact on hydrology. We agree with the likely uncertainty in the hydrological simulations due to the soil data resolution, and we agree to discuss more on the subsurface heterogeneity and data coarseness in the revision.

Comment #8: The authors have adopted manual calibration following Fatichi et al. (2015) to avoid computational load and time. This is justifiable, but manual calibration may be inefficient and subjective when dealing with many parameters across modules. Please clarify how the authors overcame this and what standardized approaches were used.

Answer: We tackle this task by calibrating the model from top to bottom and module by module. The sequential calibration of module by module is a commonly adopted logical procedure to calibrate such a fully-distributed physics-based hydrological model. We first perform manual sensitivity tests on the key parameters in each module. We then perturb and calibrate the identified sensitive parameters manually in detail to the observed variables.

The module by module calibration offers a good diagnostic power, as it isolates which modules (e.g., snowmelt, glacier melt) are causing discrepancies between observation and simulation. This allows an incremental validation, as each module can be tested and validated before integrating with the next. By doing so, errors in specific processes (e.g., snow or glacier melt) can be addressed without compromising other well-performing modules. Through the simplification of the calibration strategy, the parameter interactions are reduced, which leads to more stable model results.

Despite the calibration is sequential, we rerun the whole model (including all modules) each time when a parameter in a module is perturbed, and we focus on the model performance to the observed variables of that module. For example, when we calibrate a parameter in the snow module, we run the whole hydrological model and all temporal and spatial hydroclimatic outputs are produced, but we focus on the model performance compared to the snow water equivalent at the observed stations and spatial snow coverage. In this way, the hydrological processes between the modules are interconnected and the consistency is ensured.

We assigned the default values to the insensitive parameters given in the WaSiM user manual. Besides that, some parameter values are adopted from the literature. We agree that manual calibration is not an optimal solution. However in this study, our aim is not to reproduce the exact catchment, but to understand the hydrological processes with a reasonable parameter set, and the simulated dynamics or relative changes should be reasonably consistent. We agree to articulate our calibration procedure in more detail in the revision.

Comment #9: The authors point out an 8% underestimation of the annual glacier MB. An additional explanation about this is required. Since one glacier is used for calibration, the model may not have captured the heterogeneity of glacier responses in the catchment, as glacier MB is also affected by aspect, elevation, and other non-climatic parameters.

Answer: We agree to provide more explanation of the underestimated annual glacier MB. As only one glacier's mass balance in the catchment is available, this could lead to uncertainty in simulating the glacier responses in the whole catchment. We will discuss the influential factors mentioned by the reviewers in detail, such as the aspect, elevation, and other likely factors.

Comment #10: In Figure 2(c), please also include information about the plot within the circle. Though it is mentioned in the text, I believe some explanation is also required in the caption.

Answer: Agree and we will add this information in the caption.

Comment #11: L358: Such low discharge can also be contributed by frozen streams in winter at lower elevations, apart from the reasons mentioned.

Answer: Agree and we will add this information in the text.

Comment #12: L369–370: Is it due to the use of 1.3 m as the subsurface thickness and homogeneous subsurface properties in the model? I believe the groundwater level should show a lag time in response to surface conditions. It may also be due to the model time step (i.e. daily).

Answer: In fact, this result is obtained from the observed groundwater level and river level data, The groundwater from the deep circulation could have a long lag time in response to the surface conditions. However, the shallow groundwater observed here within 1.2 m depth shows to be as responsive as the river water level to peak melts and rainfall events in the headwater subcatchment. A lag less than 24 hours cannot be shown in the daily simulation though.

Comment #13: L385–394: I believe that the delayed response, especially in the early melting period, is valid. Additionally, the statement challenging the commonly adopted hydrological modelling approach about the role of soil needs more evidence.

Answer: Here could be a misunderstanding due to the wording. In fact, the observed shallow groundwater heads show very fast response to the early melt (nearly no delay) – this is an observed phenomenon. The simulated delayed response for months is thus incorrect when the subsurface lateral flow is allowed in the model. Additionally, the simulated groundwater hydrographs with delays depart from the observed groundwater hydrographs. That is, the observed groundwater hydrographs can only be simulated when the subsurface lateral flow is forced to 0 in this study. This finding could relate to site characteristics. We will look for more evidence on supporting this statement.

Comment #14: Section 5.6 can be made shorter; I see several repetitions of points already mentioned earlier.

Answer: We will try to shorten the length of this section. The points that have been discussed following the results will be shortened in this section.

**Reference**

Schulla, J.: Model Description WaSiM (Water balance Simulation Model), http://www.wasim.ch/downloads/doku/wasim/wasim_2024_en.pdf, 2024.